# Multimorbidity patterns in the working age population with the top 10% medical cost from exhaustive insurance claims data of Japan Health Insurance Association

Yuki Nishida[1,2,3], Tatsuhiko Anzai[1], Kunihiko Takahashi[1], Takahide Kozuma[4], Eiichiro Kanda[5], Keita Yamauchi[2], Fuminori Katsukawa[3]*

1 Department of Biostatistics, M&D Data Science Center, Tokyo Medical and Dental University, Tokyo, Japan, 2 Graduate School of Health Management, Keio University, Yokohama, Kanagawa, Japan, 3 Sports Medicine Research Center, Keio University, Yokohama, Kanagawa, Japan, 4 Department of Internal Medicine, School of Medicine, Keio University, Tokyo, Japan, 5 Medical Science, Kawasaki Medical School, Okayama, Japan

* fuminori@keio.jp

**Data Availability Statement:** The data used for this study are third party data owned by the Japan

## Abstract

Although the economic burden of multimorbidity is a growing global challenge, the contribution of multimorbidity in patients with high medical expenses remains unclear. We aimed to clarify multimorbidity patterns that have a large impact on medical costs in the Japanese population. We conducted a cross-sectional study using health insurance claims data provided by the Japan Health Insurance Association. Latent class analysis (LCA) was used to identify multimorbidity patterns in 1,698,902 patients who had the top 10% of total medical costs in 2015. The present parameters of the LCA model included 68 disease labels that were frequent among this population. Moreover, subgroup analysis was performed using a generalized linear model (GLM) to assess the factors influencing annual medical cost and 5-year mortality. As a result of obtaining 30 latent classes, the kidney disease class required the most expensive cost per capita, while the highest portion (28.6%) of the total medical cost was spent on metabolic syndrome (MetS) classes, which were characterized by hypertension, dyslipidemia, and type 2 diabetes. GLM applied to patients with MetS classes showed that cardiovascular diseases or complex conditions, including malignancies, were powerful determinants of medical cost and mortality. MetS was classified into 7 classes based on real-world data and accounts for a large portion of the total medical costs. MetS classes with cardiovascular diseases or complex conditions, including malignancies, have a significant impact on medical costs and mortality.

## Introduction

Japan implements a universal medical care insurance system in which all citizens subscribe to healthcare insurance systems. People in the working-age population (under the age of 70) receive treatment at 30% of the total direct medical costs. The remaining 70% is covered by the employee's insurance premium, of which the employee and employer each pay 50% [1].

Health Insurance Association (https://www.kyoukaikenpo.or.jp.e.ame.hp.transer.com/), and publicly unavailable because of including personal information. Data are available from the Japan Health Insurance Association (https://www.kyoukaikenpo.or.jp.e.ame.hp.transer.com/g7/cat740/sb7210/20210401/) for those study groups that apply for a competitive research grant and receive funding. The authors confirm that the authors did not have any special access or request privileges that other researchers would not have. Contact information is below: Email 99kenkyu.86t@kyoukaikenpo.or.jp.

**Funding:** This study was supported by the Japan Health Insurance Association. The funders provided the present database and had no role in the study design, data collection and analysis, decision to publish, or preparation of the manuscript.

**Competing interests:** Fuminori Katsukawa received funding from the Japan Health Insurance Association. Yuki Nishida was paid from funding from the Japan Health Insurance Association for 1 year from April 2021 to March 2022. The other authors declare no competing interests. This does not alter our adherence to PLOS ONE policies on sharing data and materials.

**Abbreviations:** AIC, Akaike information criterion; BIC, Bayesian information criterion; CKD, chronic kidney disease; GLM, generalized linear model; ICD10, International Statistical Classification of Diseases and Related Health Problems 10th Revision; LCA, latent class analysis; MetS, metabolic syndrome; NAFLD, non-alcoholic fatty liver disease; SES, socioeconomic status.

Furthermore, medical expenses incurred during treatment are set at a uniform price nationwide, and patients can freely access any health care provider. This greatly contributes to the stabilization of people's lives and maintenance of their health. However, it is difficult to maintain this system because of the imbalance between the increase in medical expenditures and the low economic growth caused by an aged society with fewer children. Prolonging this situation will force insurance fees to increase, placing a financial burden on subscribers and companies. In the worst-case scenario, the medical insurance system might go bankrupt because of an increase in dismissals and unemployment. Therefore, to sustain the medical insurance system, the growth of high medical expenses must be curtailed.

Recently, many developed countries have entered an aging society and have attempted to control health care costs for elderly people. Japan is one of the countries with the highest aging population, and there are several studies on health care costs focusing on the elderly population [2, 3]. Total medical expenses among the working-age population (younger than 65 years of age) account for approximately 40%, and these costs are expected to increase due to an increase in noncommunicable diseases [4]; thus, health care cost containment among the working-age population is also an important challenge to sustain the medical insurance system.

A small proportion of patients in the United States account for the majority of total medical expenditures [5]. Similarly, the top 10% of patients account for 60% of total annual health care costs in the working-age population in Japan [6]. If patients with mild symptoms are prevented from moving to the high-cost population, it might be possible to curb future increases in medical costs. The high-cost population is expected to include not only patients with a rare disease requiring an expensive drug or surgery due to an unexpected accident, but also those who have multiple chronic conditions, in other words, "multimorbidity" [7].

Multimorbidity is defined as two or more coexisting chronic conditions in an individual [8], and it is a growing global challenge with substantial effects on individuals, caregivers, and society. This condition affects health outcomes (e.g., increased mortality or deterioration of physical function and quality of life), patient burden (e.g., increase in the number of consultations or risk of polypharmacy), and health economic burden (e.g., increased medical expenses due to emergency hospitalization and increased consultation opportunities) [9, 10]. Although a single disease-oriented model is not suitable for the treatment of multimorbidity [11], patients in Japan are often prone to fragmented treatment. To develop a new treatment model for multimorbidity, it is important to clarify multimorbidity patterns in Japanese individuals. A recent study emphasized the need for primary prevention in the younger population because the risk of mortality was higher in younger and middle-aged adult populations with multimorbidity than in older populations [12]. Therefore, clarifying multimorbidity patterns among the working-age population with high medical costs would help health managers and policymakers to curb the increase in future medical expenses and patients with critical diseases.

The present study aimed to clarify the contribution of multimorbidity among the working-age population with the highest medical cost using latent class analysis (LCA) and the patterns that have a large impact on medical expenses. We also analyzed exhaustive insurance claims data from the Japan Health Insurance Association and provided a greater understanding of multimorbidity in high-cost patients.

## Methods

### Ethics approval

Anonymized data were used in this study. This study (S1 Checklist) was approved by the Ethics Committees of Keio University (approval number [no.]: 2020–05) and Tokyo Medical and

**Table 1. Sixty-eight disease labels classified from the ICD10 codes.**

| No. | Disease labels | ICD10 code |
|---|---|---|
| 1 | Tuberculosis of lung | A16 |
| 2 | Viral hepatitis | B16, B18, B19 |
| 3 | Other infectious diseases | A04, A09, A41, A49, A53, A56, B00, B02, B07, B35, B37, B48 |
| 4 | Gastric cancer | C16 |
| 5 | Colorectal cancer | C18, C20 |
| 6 | Cirrhosis / Liver failure / Liver cancer | C22, K72, K74 |
| 7 | Lung cancer | C34 |
| 8 | Breast cancer | C50 |
| 9 | Cervical cancer | C53 |
| 10 | Endometrial cancer | C54 |
| 11 | Ovarian cancer | C56 |
| 12 | Prostate cancer | C61 |
| 13 | Other malignancy | C25, C67, C73, C77, C78, C79, C85 |
| 14 | Benign tumor | D18, D25, D27, D37, D38, D39, D41, D43, D44, D48 |
| 15 | Iron deficiency anemia | D50 |
| 16 | Blood disease | D64, D65, D68, D69, D70 |
| 17 | Type 2 diabetes | E11, E14 |
| 18 | Dyslipidemia | E78 |
| 19 | Hyperuricemia/Gout | E79, M10 |
| 20 | Other endocrine, nutritional and metabolic diseases | E03, E04, E05, E06, E07, E10, E21, E22, E23, E28, E53, E56, E66, E83, E86, E87, E88 |
| 21 | Schizophrenia | F20 |
| 22 | Mood disorder | F31, F32 |
| 23 | Neurotic disorder | F41, F45, F48 |
| 24 | Epilepsy | G40 |
| 25 | Sleep disorder | G47 |
| 26 | Peripheral neuropathy | G56, G62, G64 |
| 27 | Autonomic neuropathy | G90 |
| 28 | Other disorders of nervous system | G98 |
| 29 | Eye disease | H00, H01, H02, H04, H10, H11, H16, H18, H20, H25, H26, H33, H35, H40, H43, H52, H53 |
| 30 | Ear disease | H60, H61, H65, H66, H68, H81, H90, H91, H93 |
| 31 | Hypertensive disease | I10, I11 |
| 32 | Ischemic heart disease | I20, I21, I25 |
| 33 | Arrhythmia | I47, I48, I49 |
| 34 | Heart failure | I50 |
| 35 | Cerebrovascular diseases | I61, I63, I65, I67, I69 |
| 36 | Arteriosclerosis | I70, I71, I73, I74 |
| 37 | Phlebitis and thrombophlebitis | I80 |
| 38 | Other cardiovascular diseases | I34, I38, I51, I95 |
| 39 | Chronic inflammation of the upper and lower airways | J30, J31, J32, J34, J37, J38, J40 |
| 40 | Chronic obstructive pulmonary disease | J42, J43, J44 |
| 41 | Asthma | J45 |
| 42 | Interstitial pneumonia | J84 |
| 43 | Oral disease | K02, K05, K03, K04, K06, K07, K08, K12 |
| 44 | Reflux esophagitis | K21 |
| 45 | Gastric ulcer/Duodenal ulcer | K25, K26 |

*(Continued)*

**Table 1.** (Continued)

| No. | Disease labels | ICD10 code |
|---|---|---|
| 46 | Gastritis | K29 |
| 47 | Ulcerative colitis | K51 |
| 48 | Ileus | K56 |
| 49 | Hemorrhoid | I84 |
| 50 | Alcoholic liver disease | K70 |
| 51 | Other liver diseases | K76, K73 |
| 52 | Gallbladder/Biliary tract disease | K80, K81, K82 |
| 53 | Acute pancreatitis | K85 |
| 54 | Other gastrointestinal disorders | K30, K31, K52, K57, K58, K59, K60, K62, K63, K75, K86, K92 |
| 55 | Skin and subcutaneous tissue | L01, L02, L03, L08, L20, L21, L25, L27, L29, L30, L40, L50, L70, L72, L73, L81, L84, L85, L91, L98, Q82 |
| 56 | Connective tissue disease | M06, M32, M35 |
| 57 | Chronic disease of load-bearing joints | M16, M17, M47, M51 |
| 58 | Osteoporosis | M81 |
| 59 | Other locomotor disorders | M19, M13, M25, M43, M48, M50, M53, M54, M62, M65, M72, M75, M77, M79 |
| 60 | Chronic kidney disease | N18 |
| 61 | Other kidney diseases | N05, N12, N19, N28, Q61 |
| 62 | Calculus of kidney and ureter | N20 |
| 63 | Other urinary disorders | N30, N31, N13, N32, N39 |
| 64 | Male genital disorders | N40, N41 |
| 65 | Inflammatory diseases of female pelvic organs | N72, N76 |
| 66 | Noninflammatory disorders of female genital tract | N80, N92, N93, N94, N95, N97, N85, N86, N87 |
| 67 | Hemorrhage in early pregnancy | O20 |
| 68 | Breast disorders | N63, N64 |

ICD10, International Statistical Classification of Diseases and Related Health Problems 10th Revision.

Dental University (approval no.: M2020-385) and was exempt from the need to obtain informed consent from participants.

## Study design and data source

We conducted a cross-sectional study using exhaustive health insurance claims data provided by the Japan Health Insurance Association, which comprises the largest working-age population in Japan. This database consists of approximately 40 million people younger than 75 years of age who are formally or informally employed at small to medium enterprises and their family members. The claim data include information on total direct medical costs including patients' co-payment comprised from inpatient services, outpatient services, and outpatient prescription drugs. However, socioeconomic status (SES), such as education level and income, are not included. Currently, only academic researchers selected by open recruitment can use data for the 7 years from 2015 to 2021.

Diagnosed disease prevalence was represented by the International Statistical Classification of Diseases and Related Health Problems 10th Revision (ICD10), which was documented between April 2015 and March 2016. The annual medical cost for each patient was calculated

by summing the outpatient, hospital, drug, and dental costs during the same period (JPY was translated to USD at the 2015 rate of 1 USD = 120 JPY).

## Study population

The study population comprised insurance affiliates aged 18–64 years as of April 1, 2015 (n = 20,803,665). The exclusion criteria were as follows: insurance affiliates younger than 18 years of age and older than 65 years of age as of April 1, 2015 (n = 1,454,866) and those who took out insurance or withdrew in the middle of the 2015 fiscal year (n = 2,359,770).

## Analyses

LCA was used to identify multimorbidity patterns in the population with high medical costs. LCA is a model-based approach to classification that permits a statistical evaluation of how well the proposed LCA model represents the data [13]. The present parameters of the LCA model were disease labels based on the ICD10 diagnosis code. ICD10 diagnosis codes were indicated with the first 3 characters, and the following codes were excluded: (1) codes, the frequency of which in the present population was less than 1%, which were excluded to obtain consistent and clinically interpretable patterns of association; (2) transient disease codes such as common cold; (3) symptom codes (R00-R99); and (4) non-disease codes (S00-U99). As a result, 252 diagnostic codes were extracted (S1 Table). Furthermore, these diagnosis codes were classified into 68 disease labels based on pathological similarity by an expert panel (Table 1), and the labels were used in LCA as a binary variable. Labels assigned to the identified latent disease classes were determined with reference to the highest item-response probability in the class, higher probability than the other classes, treatment content, and sex proportion. The item-response probabilities of the disease labels are shown in a heatmap for each class for visual clarity. The prevalence rates of some disease labels (e.g., oral disease, eye disease, chronic inflammation of the upper and lower airways, and skin and subcutaneous tissue) were high in most classes because the present study focused on patients with high medical costs, and the prevalence rates of these diseases in the present study were higher than those in the general population, including healthy individuals. Therefore, if the probability of the class was below the prevalence rate in all individuals, the box in the heatmap was represented as colorless.

Multimorbidity is formally defined as 2 or more chronic conditions [14]; thus, we selected 46 chronic conditions from 68 disease labels in Table 2 and estimated the multimorbidity (S2 Table). In chronic diseases, malignancy was grouped into one category, and the following diseases not important in terms of multimorbidity were excluded: oral disease, eye disease, and skin and subcutaneous tissue disease.

In addition, subgroup analysis was performed and focused on metabolic syndrome (MetS) classes, which were characterized by the MetS triad (type 2 diabetes, dyslipidemia, and hypertensive disease) because these medical costs accounted for the largest share of the total. The generalized linear model (GLM) with a log-link function and gamma distribution was generated to assess the factors influencing the total cost and marginal effects of the total medical cost [15]. Multiple logistic regression analysis was used to estimate the odds ratios for the associations with the 5-year mortality rate. A 95% confidence level was used to determine statistically significant variables for the total cost and mortality.

Representative values are shown as medians [quartile Q1, Q3]. LCA was performed using JMP Pro 16 (SAS Institute Japan Co., Ltd., Cary, NC, USA). R 4.3.0 statistical software was used for the other analyses, and average marginal effect of cost was calculated using the margins package.

**Table 2. Demographic characteristics of the study population.**

| | Overall | Men | Women |
|---|---|---|---|
| n | 1,698,902 | 1,027,452 | 671,450 |
| Age (years) | | | |
| Median [Q1, Q3] | 52 [42, 59] | 54 [42, 59] | 49 [39, 57] |
| 18–29, n (%) | 92,973 (5.5) | 36,076 (3.5) | 56,897 (8.5) |
| 30–39 | 229,952 (13.5) | 109,413 (10.6) | 120,539 (18.0) |
| 40–49 | 404,092 (23.8) | 236,558 (23.0) | 167,534 (25.0) |
| 50–59 | 591,525 (34.8) | 372,174 (36.2) | 219,351 (32.7) |
| 60–64 | 380,360 (22.4) | 273,231 (26.6) | 107,129 (16.0) |
| Number of diagnosis codes out of the list of 68 disease labels (n) | | | |
| Median [Q1, Q3] | 10 [7, 14] | 10 [7, 14] | 11 [8, 15] |
| Number of co-existing chronic conditions out of the selected list of 46 chronic conditions (n) | | | |
| Median [Q1, Q3] | 7 [5,10] | 7 [5, 10] | 7 [5, 10] |
| Chronic conditions ≥2 (%) | 95.6 | 95.0 | 96.6 |
| Chronic conditions ≥3 (%) | 91.1 | 90.2 | 92.3 |
| Direct medical cost (JPY) * | | | |
| Total cost | 1,397,311,088,550 (11,644,259,071 USD) | 904,081,490,850 (7,534,012,424 USD) | 493,229,597,700 (4,110,246,648 USD) |
| Per capita | 822,480 (6,854 USD) | 879,930 (7,333 USD) | 734,570 (6,121 USD) |
| Median [Q1, Q3] | 432,940 [331,010, 717,150] (3,608 [2,758, 5,976] USD) | 436,130 [332,490, 726,850] (3,634 [2,771, 6,057] USD) | 428,170 [328,870, 703,380] (3,568 [2,741, 5,862] USD) |
| Inpatient cost (%) | 39.4 | 39.8 | 38.9 |
| Drug cost (%) | 21 | 21.2 | 20.6 |
| The number of outpatient visits (n/year) | | | |
| <12, n (%) | 215,784 (12.7) | 149,152 (14.5) | 66,632 (9.9) |
| 12–23 | 569,430 (33.5) | 370,844 (36.1) | 198,586 (29.6) |
| 24–35 | 466,763 (27.5) | 273,219 (26.6) | 193,544 (28.8) |
| ≥36 | 446,925 (26.3) | 234,237 (22.8) | 212,688 (31.7) |
| Length of hospital stay (days/years) | | | |
| <3, n (%) | 1,161,361 (68.4) | 708,765 (69.0) | 452,596 (67.4) |
| 3–7 | 190,472 (11.2) | 115,086 (11.2) | 75,386 (11.2) |
| 8–30 | 277,693 (16.3) | 158,434 (15.4) | 119,259 (17.8) |
| ≥31 | 69,376 (4.1) | 45,167 (4.4) | 24,209 (3.6) |
| 5-year retirement rate (%) | 46.1 | 44.3 | 48.8 |
| 5-year mortality rate (%) | 1.7 | 2.3 | 0.9 |

*1 USD = 120 JPY

Q, quartile.

## Results

### Demographic characteristics of participants

After excluding the inapplicable persons, we extracted the top 10% of patients with high medical costs and finally analyzed the data of 1,698,902 patients (671,450 women and 1,027,452 men). The distribution of annual medical costs in all insured populations is shown in S1 Fig. The demographic characteristics of the study population are shown in Table 2. The median

number of co-existing chronic conditions in an individual, out of the selected list of 46 chronic conditions, was 7 [5, 10]. Overall, 95.6% and 91.1% of all patients had 2 or more conditions and 3 or more conditions, respectively; thus, most of them were in a multimorbidity state. The total medical cost was 1,397,311,088,550 JPY (11,644,259,071 USD) and that per capita was 822,480 JPY (6,854 USD). Of the total costs, the proportions of inpatient cost and drug cost were 39.4% and 21.0%, respectively. Men incurred higher medical cost per capita [879,930 JPY (7,333 USD)] than women [734,570 JPY (6,121 USD)], whereas the number of outpatient visits for women seemed to be higher than that for men. The median length of hospital stay was 0 [0, 5], and 31.6% of patients were hospitalized for more than 3 days. Five-year retirement and mortality rates were 46.1% and 1.7%, respectively. The comparisons of demographic characteristics between the top 10% of patients with high medical costs and patients with low to medium costs (i.e., lower 90%) or 0 costs are shown in S3 and S4 Tables.

## Determining the number of latent classes

In the preliminary analysis, we started with a 2-class model and successively increased the number of latent models to 40 to select the optimal number of latent classes. Model fitness was examined using the Akaike information criterion (AIC) and Bayesian information criterion (BIC). However, the AIC and BIC did not reach a minimum until 40 classes and seemed to continue to decline thereafter (S2 Fig). Too many classes would make the interpretation difficult; thus, we examined 5 models of 20, 25, 30, 35, and 40 classes, and assigned labels to the identified latent disease classes in each model (S3 Fig). Common classes that appeared in all models were as follows: single chronic condition, chronic inflammation of airways, mental disease, digestive disease, motor disorder, diseases specific to women or men, MetS, connective tissue disease, malignancy, and kidney disease. Classes of urologic disease did not appear in the 20-class model, whereas some classes were difficult to interpret in the 35- and 40- class models. Moreover, malignancy classes were more clearly identified in the 30-class model than in the 25-class model. Consequently, a 30-class model was selected as the final model.

## Characteristics of the 30 latent class models

Table 3 summarizes the label names and the features of each class, and Table 4 shows characteristics of the 30 latent class models as follows: single chronic condition (6.9%); otorhinolaryngology diseases (5.8%); 3 types of mental disease (6.7%); liver disease (3.3%); 2 types of digestive diseases (7.9%); 2 types of motor disorders (7.8%); urologic disease (2.0%); 4 types of diseases specific to women (11.9%); diseases specific to men (2.6%); 7 types of MetSs (31.8%); connective tissue disease (2.2%); 5 types of malignancies (10.0%); and kidney disease (1.2%). A heatmap of all classes is shown in S4 Fig. Class 1 was named as the "single chronic condition." Patients in this class frequently underwent surgery for any fractures and had only one chronic condition (1 [1, 2]); therefore, they were considered as patients with non-multimorbidity. Mental disease classes included patients with a high prevalence of mood disorder, neurotic disorder, and sleep disorder simultaneously and suggested a higher 5-year retirement rate (51.4–54.7%) than that of other classes. If more than 95% of patients within the class were women or men, that class was named as a sex-specific disease class. One of these sex-specific classes included younger women (33 [29, 37] years) who received perinatal care, and the other classes consisted of patients with multimorbidity. There were 7 MetS classes that showed high frequencies of type 2 diabetes, dyslipidemia, and hypertension simultaneously. Malignancy classes were classified into breast cancer, lung cancer, cirrhosis/hepatoma, gastrointestinal cancer including stomach cancer or colorectal cancer, and cancer except for genital cancers. Although the breast cancer class consisted mostly of females (98.3%), the prevalence of female genital

**Table 3. Features of the 30 latent class models.**

| Class No. | Assigned label | Features |
|---|---|---|
| 1 | Single chronic condition | ・Patients had only one chronic condition. |
| 2 | Otorhinolaryngology disease | ・95% of patients had chronic inflammation of the upper and lower airways. |
| | | ・The prevalence of ear disease was higher than that in the other classes. |
| 3 | Mental disease 1 | ・Mood disorder and sleep disorder were conspicuous. |
| | | ・The prevalence of schizophrenia and neurotic disorder were higher than that in the other classes. |
| 4 | Mental disease 2 | ・Mood disorder and sleep disorder were conspicuous. |
| | | ・Metabolic disorders such as type 2 diabetes and dyslipidemia were also conspicuous. |
| 5 | Mental disease 3 | ・Mood disorder and sleep disorder were conspicuous. |
| | | ・The proportion of females (78.5%) was higher than that in other mental classes. |
| 6 | Liver disease | ・92% of patients had other liver diseases. |
| | | ・Metabolic disorders such as type 2 diabetes and dyslipidemia were also conspicuous. |
| 7 | Digestive disease 1 | ・The prevalence of other gastrointestinal disorders was the highest. |
| | | ・40% of patients had colorectal cancer. |
| 8 | Digestive disease 2 | ・The prevalence of other gastrointestinal disorders was the highest. |
| | | ・41% and 33% of patients had colorectal cancer and gastric cancer, respectively. |
| | | ・The number of chronic conditions was higher than that of another digestive disease class. |
| 9 | Motor disorders 1 | ・81% and 88% of patients had chronic disease of load-bearing joints and other locomotive disease, respectively. |
| 10 | Motor disorders 2 | ・66% and 96% of patients had chronic disease of load-bearing joints and other locomotive disease, respectively. |
| | | ・The number of conditions was higher than that of another motor disorders class. |
| 11 | Urologic disease | ・Other urinary disorders was conspicuous, and their prevalence was the highest in all classes. |
| 12 | Diseases specific to women 1 | ・97.8% of patients were female. |
| | | ・Age is the youngest in all the classes, and 50% of patients had hemorrhage in early pregnancy. |
| 13 | Diseases specific to women 2 | ・99.2% of patients were female. |
| | | ・84% of patients had noninflammatory disorders of the female genital tract. |
| | | ・37% and 34% of patients had endometrial cancer and ovarian cancer, respectively. |
| 14 | Diseases specific to women 3 | ・99.5% of patients were female. |
| | | ・83% of patients had noninflammatory disorders of the female genital tract. |
| | | ・33% of patients had endometrial cancer. |
| | | ・Metabolic disorders such as type 2 diabetes and dyslipidemia were also conspicuous. |
| 15 | Diseases specific to women 4 | ・99.6% of patients were female. |
| | | ・87% of patients had noninflammatory disorders of the female genital tract. |
| 16 | Diseases specific to men | ・99.9% of patients were male. |
| | | ・77% of patients had prostate cancer. |
| | | ・The prevalence of metabolic triad was moderately high. |
| | | ・91% of patients had male genital disorders. |
| 17 | MetS 1 | ・The metabolic triad* was conspicuous. |
| 18 | MetS 2 | ・The metabolic triad* was conspicuous. |
| | | ・The prevalences of infection, chronic inflammation of the upper and lower airways, oral disease, gastrointestinal tract disease, and other locomotive diseases were moderately high. |
| 19 | MetS 3 | ・The metabolic triad* was conspicuous. |
| | | ・87% and 96% of patients had chronic disease of load-bearing joints and other locomotive disease, respectively. |
| 20 | MetS 4 | ・The metabolic triad* was conspicuous. |
| | | ・70% and 49% of patients had other liver diseases and chronic kidney disease, respectively. |
| 21 | MetS 5 | ・The metabolic triad* was conspicuous. |
| | | ・Cardiovascular diseases were conspicuous. |
| 22 | MetS 6 | ・The metabolic triad* was conspicuous. |
| | | ・Cardiovascular diseases, other metabolic diseases, gastrointestinal disease, and other locomotive diseases were conspicuous. |

(*Continued*)

**Table 3.** (Continued)

| Class No. | Assigned label | Features |
|---|---|---|
| 23 | MetS 7 | ・The metabolic triad* was conspicuous. |
| | | ・The number of chronic conditions was highest in all classes. |
| 24 | Connective tissue disease | ・96% of patients had connective tissue disease. |
| 25 | Breast cancer | ・94% of patients had breast cancer. |
| | | ・98.3% of patients were female, and the female genital tract was not conspicuous when compared with diseases specific to the female classes. |
| 26 | Lung cancer | ・The prevalence of lung disease was the highest in all classes. |
| 27 | Cirrhosis/Liver cancer | ・The prevalence of cirrhosis/liver failure/liver cancer was the highest in all classes. |
| 28 | Gastrointestinal cancer | ・The prevalences of gastric cancer and colorectal cancer were highest in all classes. |
| 29 | Cancer except for genital cancer | ・The prevalence of cancer except for genital cancer was moderately high. |
| 30 | Kidney disease | ・90% and 99% of patients had chronic kidney disease and other kidney diseases, respectively. |

*Metabolic triad means type 2 diabetes, dyslipidemia, and hypertensive disease.

MetS, metabolic syndrome.

tract cancer was not substantial and even males can develop breast cancer infrequently; thus, the breast cancer class was separated from the sex-specific disease class. Patients in these malignancy classes showed a higher 5-year mortality rate than those in the other classes. The kidney disease class had the smallest number of patients in all classes and showed that the prevalence rates of chronic kidney disease (CKD) and hypertension were 90% and 95%, respectively.

## Latent class ratios by age and sex

Fig 1 shows the percentage of each patient per class as a 100% stacked vertical bar chart by age group and sex. Among men, the single chronic condition class included a large proportion of patients aged 18–29 years, and the percentage of patients in the MetS classes increased with age and reached more than 50% after age 50–59 years. Among women, patients with diseases specific to their sex accounted for nearly half for those aged 18–29 and 30–39 years. The percentages of breast cancer, MetS, and motor disorder classes increased from age 40s to 60s. As a feature common to both sexes, mental disease classes were larger in the young population than in the old population.

## The total medical cost and cost per capita by latent classes

The total cost and cost per person for each class are shown in Fig 2. Although total cost tended to be higher in the class with a large number of patients, some classes of MetS, i.e., the malignancy class and kidney disease class, showed higher total cost due to a high cost per capita. In particular, the kidney disease class showed the highest total cost and cost per person among all classes, despite the small number of patients.

## Subgroup analysis of the seven MetS classes

Herein, 31.8% of patients were classified into some type of MetS class, and their combined medical costs accounted for 28.6% of the total. To understand the characteristics of the MetS class in more detail, a heatmap focusing on the 7 MetS classes is shown in Fig 3, in which only highly frequent diseases in the MetS class are shown. The notable features of each MetS class

**Table 4. Characteristics of the 30 latent class models.**

| Class no. | Assigned label | n | Age years | Female % | Number of diagnosis codes out of the list of 68 disease labels n | Number of co-existing chronic conditions out of the selected list of 46 chronic conditions n | Inpatient costs % | Drug costs % | The number of outpatient visits n/year | Length of hospital stay days/ years | 5-year retirement rate % | 5-year mortality rate % |
|---|---|---|---|---|---|---|---|---|---|---|---|---|
| 1 | Single chronic condition | 116,940 | 43 [34, 53] | 32.0 | 3 [2, 4] | 1 [1, 2] | 50.2 | 14.4 | 16 [9,27] | 2 [0,8] | 41.6 | 0.6 |
| 2 | Otorhinolaryngology disease | 98,369 | 46 [37, 55] | 48.9 | 8 [6, 10] | 5 [3, 6] | 29.4 | 26.1 | 28 [19,39] | 0 [0,3] | 43.8 | 0.5 |
| 3 | Mental disease 1 | 42,350 | 43 [35, 50] | 33.1 | 7 [5, 8] | 5 [4, 6] | 27.5 | 34.3 | 27 [19,36] | 0 [0,0] | 51.4 | 1.0 |
| 4 | Mental disease 2 | 41,080 | 48 [41, 55] | 29.7 | 13 [11, 16] | 10 [9, 12] | 35.4 | 28.7 | 30 [22,41] | 0 [0,0] | 52.6 | 1.9 |
| 5 | Mental disease 3 | 29,697 | 43 [35, 50] | 78.5 | 13 [12, 16] | 9 [8, 11] | 24.2 | 31.9 | 36 [27,47] | 0 [0,0] | 54.7 | 0.8 |
| 6 | Liver disease | 55,869 | 53 [44, 59] | 42.5 | 13 [12, 15] | 9 [8, 11] | 25.3 | 28.7 | 28 [20,38] | 0 [0,0] | 44.5 | 0.9 |
| 7 | Digestive disease 1 | 80,754 | 48 [39, 57] | 26.8 | 7 [6, 9] | 4 [3, 6] | 47.8 | 17.1 | 16 [9,24] | 3 [0,9] | 42.0 | 1.8 |
| 8 | Digestive disease 2 | 53,326 | 52 [43, 58] | 47.2 | 14 [12, 16] | 9 [8, 11] | 35.0 | 20.6 | 31 [22,42] | 0 [0,4] | 45.8 | 2.0 |
| 9 | Motor disorders 1 | 70,368 | 52 [44, 59] | 39.9 | 8 [6, 9] | 5 [4, 6] | 48.7 | 16.1 | 29 [18,45] | 0 [0,7] | 47.3 | 0.7 |
| 10 | Motor disorders 2 | 62,165 | 55 [49, 60] | 63.2 | 16 [14, 18] | 12 [10, 13] | 26.9 | 27.8 | 37 [27,52] | 0 [0,0] | 49.7 | 0.9 |
| 11 | Urologic disease | 33,215 | 55 [46, 61] | 11.1 | 8 [6, 10] | 5 [4, 7] | 47.1 | 15.0 | 20 [13,30] | 2 [0,6] | 46.0 | 1.3 |
| 12 | Diseases specific to women 1 | 52,066 | 33 [29, 37] | 97.9 | 8 [6, 10] | 4 [3, 6] | 73.2 | 5.8 | 18 [12,28] | 9 [3,13] | 46.5 | 0.1 |
| 13 | Diseases specific to women 2 | 51,107 | 44 [37, 49] | 99.2 | 10 [8, 12] | 7 [5, 8] | 58.3 | 11.0 | 20 [13,29] | 4 [0,9] | 42.3 | 0.9 |
| 14 | Diseases specific to women 3 | 50,179 | 49 [42, 55] | 99.5 | 20 [18, 22] | 14 [12, 16] | 38.7 | 18.6 | 36 [26,49] | 0 [0,6] | 46.9 | 1.6 |
| 15 | Diseases specific to women 4 | 49,395 | 44 [36, 51] | 99.6 | 13 [12, 15] | 8 [7, 10] | 28.5 | 22.6 | 34 [26,46] | 0 [0,2] | 46.2 | 0.3 |
| 16 | Diseases specific to men | 44,130 | 60 [55, 63] | 0.1 | 15 [13, 18] | 11 [9, 13] | 30.6 | 24.6 | 31 [23,42] | 0 [0,2] | 50.6 | 1.6 |

(*Continued*)

**Table 4.** (Continued)

| Class no. | Assigned label | n | Age years | Female % | Number of diagnosis codes out of the list of 68 disease labels n | Number of co-existing chronic conditions out of the selected list of 46 chronic conditions n | Inpatient costs % | Drug costs % | The number of outpatient visits n/year | Length of hospital stay days/years | 5-year retirement rate % | 5-year mortality rate % |
|---|---|---|---|---|---|---|---|---|---|---|---|---|
| 17 | MetS 1 | 157,489 | 54 [47, 60] | 19.2 | 6 [5, 8] | 5 [4, 6] | 22.1 | 30.8 | 21 [14,29] | 0 [0,0] | 44.2 | 1.1 |
| 18 | MetS 2 | 102,023 | 57 [51, 61] | 18.1 | 11 [10, 13] | 8 [7, 10] | 24.8 | 29.1 | 26 [19,35] | 0 [0,0] | 46.1 | 1.0 |
| 19 | MetS 3 | 63,365 | 57 [52, 61] | 27.4 | 12 [10, 14] | 9 [8, 11] | 35.5 | 25.3 | 28 [20,41] | 0 [0,0] | 49.9 | 1.2 |
| 20 | MetS 4 | 45,267 | 56 [49, 61] | 13.6 | 11 [9, 13] | 9 [7, 10] | 30.7 | 28.7 | 19 [13,27] | 0 [0,2] | 46.0 | 1.9 |
| 21 | MetS 5 | 78,118 | 56 [49, 61] | 11.6 | 10 [8, 11] | 8 [7, 9] | 56.2 | 19.9 | 17 [11,24] | 0 [0,4] | 46.3 | 1.9 |
| 22 | MetS 6 | 61,373 | 57 [51, 61] | 17.9 | 17 [15, 19] | 13 [12, 15] | 59.3 | 16.5 | 25 [17,34] | 0 [0,9] | 48.3 | 2.6 |
| 23 | MetS 7 | 32,980 | 57 [50, 61] | 23.0 | 23 [21,25] | 17 [15, 19] | 47.6 | 17.7 | 39 [28,56] | 0 [0,11] | 48.3 | 4.5 |
| 24 | Connective tissue disease | 38,048 | 51 [42, 58] | 64.5 | 11 [9, 14] | 8 [6, 10] | 18.8 | 30.8 | 22 [15,32] | 0 [0,0] | 46.2 | 1.2 |
| 25 | Breast cancer | 29,675 | 51 [45, 57] | 98.3 | 10 [8, 13] | 7 [5, 9] | 31.8 | 12.3 | 25 [17,37] | 0 [0,7] | 44.0 | 3.5 |
| 26 | Lung cancer | 23,871 | 55 [46, 61] | 27.8 | 12 [9, 14] | 8 [6, 10] | 51.2 | 14.2 | 22 [14,31] | 3 [0,12] | 46.5 | 6.5 |
| 27 | Cirrhosis/Liver cancer | 31,736 | 51 [43, 58] | 22.9 | 9 [7, 11] | 6 [4, 7] | 18.4 | 46.2 | 18 [11,27] | 0 [0,5] | 42.6 | 2.8 |
| 28 | Gastrointestinal cancer | 60,542 | 56 [49, 61] | 20.0 | 15 [13, 17] | 10 [8, 12] | 55.1 | 13.6 | 22 [15,31] | 4 [0,15] | 46.2 | 6.6 |
| 29 | Cancer except for genital cancer | 23,521 | 54 [45, 60] | 47.5 | 21 [19, 23] | 15 [13, 17] | 54.8 | 14.7 | 30 [21,43] | 5 [0,26] | 48.0 | 8.0 |
| 30 | Kidney disease | 19,884 | 53 [46, 59] | 16.4 | 18 [15, 21] | 14 [12, 17] | 15.7 | 13.3 | 118 [24,164] | 0 [0,10] | 44.6 | 6.7 |

Median [Q1, Q3]

Q, quartile; no., number; MetS, metabolic syndrome.

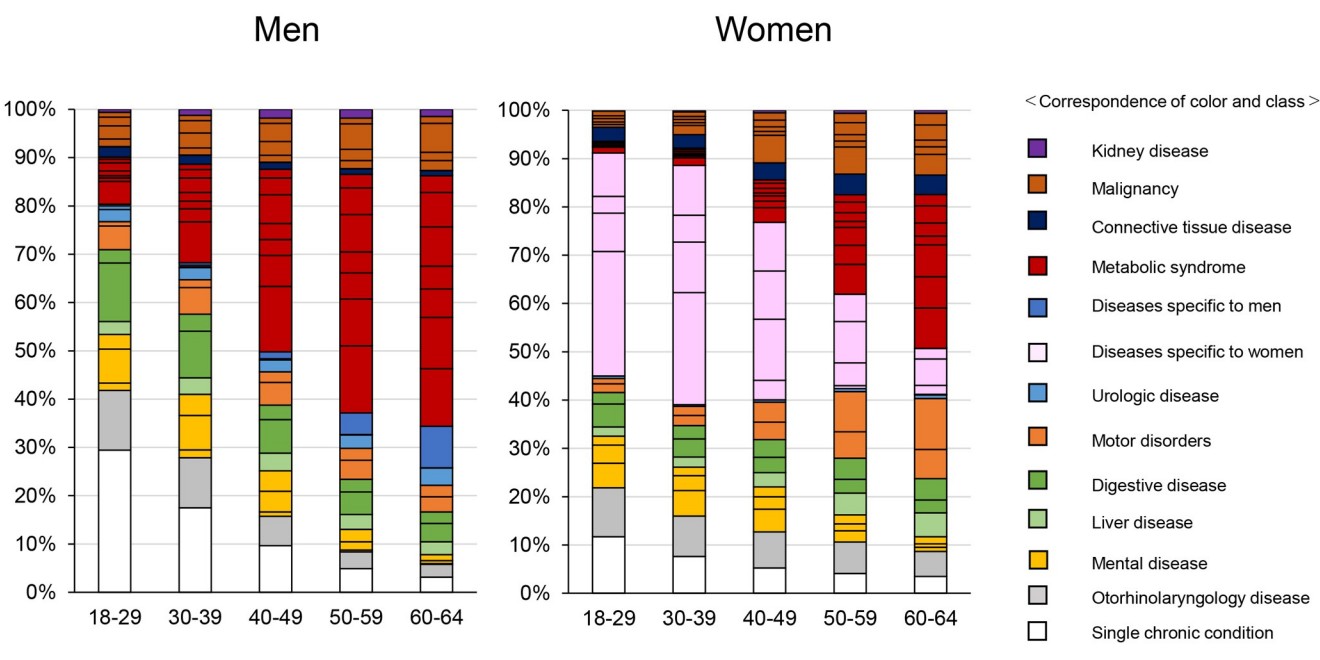

**Fig 1. A 100% stacked vertical bar of the patient count in each class by age and sex.**

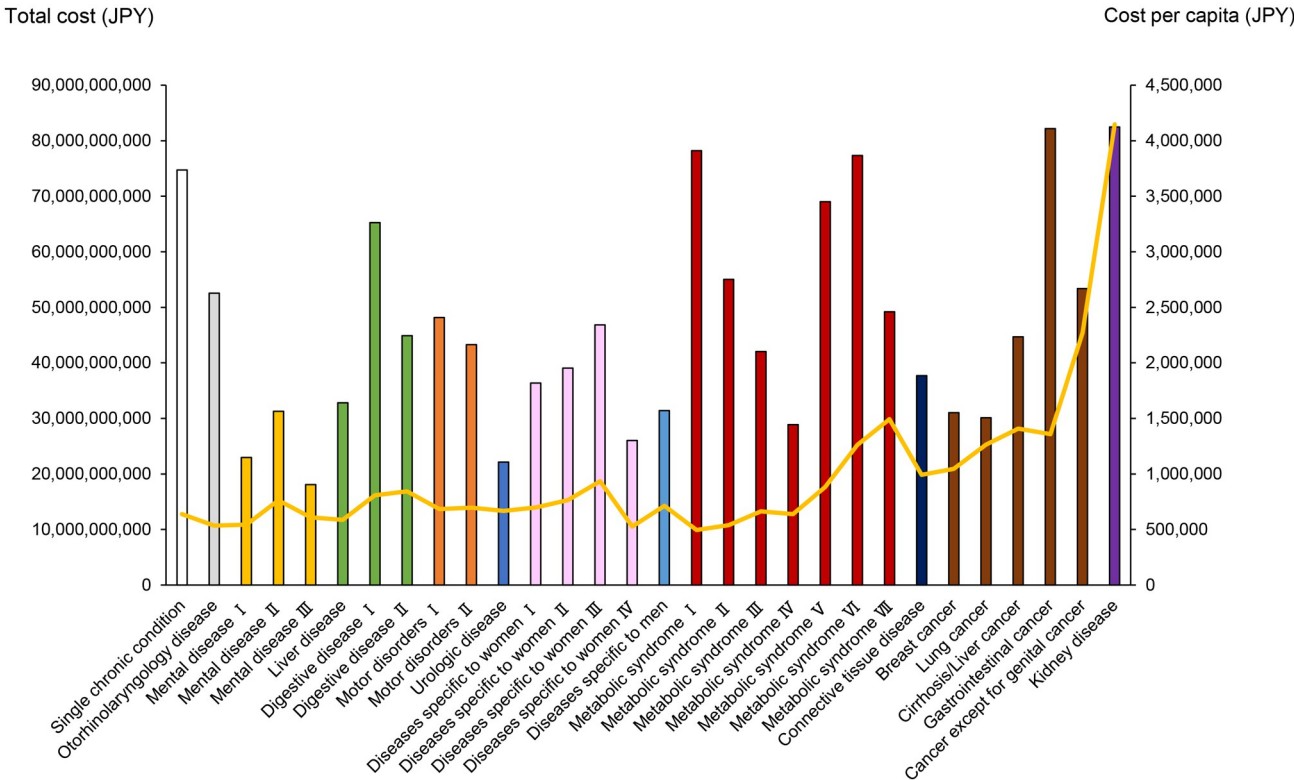

**Fig 2. Total cost and cost per capita in each class.** Vertical bars represent the total cost, and the yellow line represents the cost per capita. 1 USD = 120 JPY.

| | MetS class 1 | MetS class 2 | MetS class 3 | MetS class 4 | MetS class 5 | MetS class 6 | MetS class 7 |
|---|---|---|---|---|---|---|---|
| Other infectious diseases | 0.15 | 0.51 | 0.28 | 0.16 | 0.17 | 0.49 | 0.80 |
| Gastric cancer | 0.00 | 0.17 | 0.03 | 0.05 | 0.01 | 0.09 | 0.33 |
| Colorectal cancer | 0.01 | 0.16 | 0.05 | 0.16 | 0.02 | 0.13 | 0.39 |
| Other malignancy | 0.02 | 0.05 | 0.05 | 0.24 | 0.01 | 0.12 | 0.38 |
| Benign tumor | 0.05 | 0.11 | 0.12 | 0.15 | 0.05 | 0.19 | 0.39 |
| Blood disease | 0.04 | 0.02 | 0.12 | 0.26 | 0.15 | 0.39 | 0.43 |
| Type 2 diabetes | 0.80 | 0.65 | 0.77 | 0.93 | 0.75 | 0.90 | 0.88 |
| Dyslipidemia | 0.76 | 0.82 | 0.79 | 0.90 | 0.72 | 0.81 | 0.81 |
| Hyperuricemia / Gout | 0.21 | 0.36 | 0.32 | 0.49 | 0.23 | 0.36 | 0.42 |
| Other endocrine, nutritional and metabolic diseases | 0.16 | 0.12 | 0.25 | 0.44 | 0.28 | 0.62 | 0.72 |
| Mood disorder | 0.02 | 0.03 | 0.03 | 0.00 | 0.01 | 0.05 | 0.33 |
| Neurotic disorder | 0.03 | 0.11 | 0.07 | 0.02 | 0.05 | 0.17 | 0.49 |
| Sleep disorder | 0.13 | 0.29 | 0.20 | 0.10 | 0.16 | 0.36 | 0.69 |
| Peripheral neuropathy | 0.04 | 0.07 | 0.46 | 0.03 | 0.03 | 0.16 | 0.47 |
| Eye disease | 0.42 | 0.47 | 0.37 | 0.41 | 0.26 | 0.43 | 0.61 |
| Ear disease | 0.05 | 0.13 | 0.09 | 0.03 | 0.05 | 0.14 | 0.31 |
| Hypertensive disease | 0.68 | 0.83 | 0.79 | 0.82 | 0.83 | 0.87 | 0.72 |
| Ischemic heart disease | 0.07 | 0.18 | 0.23 | 0.28 | 0.71 | 0.83 | 0.48 |
| Arrhythmia | 0.05 | 0.09 | 0.13 | 0.13 | 0.43 | 0.55 | 0.34 |
| Heart failure | 0.02 | 0.05 | 0.14 | 0.21 | 0.73 | 0.85 | 0.36 |
| Cerebrovascular diseases | 0.14 | 0.20 | 0.26 | 0.27 | 0.25 | 0.42 | 0.42 |
| Arteriosclerosis | 0.09 | 0.12 | 0.24 | 0.27 | 0.29 | 0.49 | 0.34 |
| Other cardiovascular diseases | 0.02 | 0.03 | 0.09 | 0.11 | 0.32 | 0.49 | 0.23 |
| Chronic inflammation of the upper and lower airways | 0.27 | 0.57 | 0.31 | 0.17 | 0.23 | 0.47 | 0.73 |
| Asthma | 0.08 | 0.21 | 0.09 | 0.04 | 0.08 | 0.20 | 0.34 |
| Oral disease | 0.54 | 0.65 | 0.53 | 0.45 | 0.46 | 0.54 | 0.70 |
| Reflux esophagitis | 0.07 | 0.54 | 0.28 | 0.16 | 0.48 | 0.61 | 0.78 |
| Gastric ulcer / Duodenal ulcer | 0.05 | 0.33 | 0.19 | 0.12 | 0.22 | 0.37 | 0.58 |
| Gastritis | 0.13 | 0.69 | 0.53 | 0.19 | 0.19 | 0.51 | 0.87 |
| Other liver diseases | 0.33 | 0.38 | 0.47 | 0.70 | 0.19 | 0.49 | 0.76 |
| Gallbladder/Biliary tract disease | 0.03 | 0.11 | 0.07 | 0.13 | 0.03 | 0.12 | 0.30 |
| Other gastrointestinal disorders | 0.09 | 0.49 | 0.25 | 0.31 | 0.16 | 0.47 | 0.86 |
| Skin and subcutaneous tissue | 0.23 | 0.45 | 0.35 | 0.17 | 0.21 | 0.45 | 0.70 |
| Chronic disease of load-bearing joints | 0.08 | 0.18 | 0.87 | 0.04 | 0.08 | 0.27 | 0.56 |
| Other locomotor disorders | 0.25 | 0.55 | 0.96 | 0.26 | 0.25 | 0.62 | 0.92 |
| Other kidney diseases | 0.08 | 0.05 | 0.18 | 0.49 | 0.08 | 0.28 | 0.44 |
| Other urinary disorders | 0.04 | 0.04 | 0.11 | 0.21 | 0.05 | 0.19 | 0.38 |

☐ p < 0.25;  ☐ 0.25 ≤ p < 0.5;  ☐ 0.5 ≤ p < 0.75;  ☐ 0.75 ≤ p

**Fig 3. A heatmap for the item-response probabilities of diseases categorized in the metabolic syndrome classes.** Thick red represents a high probability.

were as follows: MetS class 1, only MetS triad (type 2 diabetes, dyslipidemia, and hypertensive disease); MetS class 2, triad plus infection, chronic inflammation of the upper and lower airways, oral disease, gastrointestinal tract disease, and other locomotive diseases; MetS class 3, triad plus motor disorder; MetS class 4, triad plus liver disease and kidney disease; MetS class 5, triad plus cardiovascular diseases; MetS class 6, triad plus other metabolic diseases, cardiovascular diseases, gastrointestinal disease, and other locomotive diseases; and MetS class 7, triad plus complex conditions including malignancies.

Table 5 shows the results of the GLM used to determine the variables affecting the annual total medical cost and 5-year mortality in the MetS classes. Total medical costs in MetS classes 5 and 6 were more than 20% higher than that in MetS class 1. The highest category in the total number of diagnosis codes out of the list of 68 disease labels (more than 30) increased the total medical cost by 1.66 times compared with the lowest category (less than 10), and its average marginal effect (466,361 yen) was larger than that of any MetS class. The 5-year mortality rate in MetS class 7 was the highest of all classes (odds ratio = 2.47).

## Discussion

This study is the first attempt to clarify multimorbidity patterns among working-age patients with high medical costs using LCA. The strength of our study is that the present results of multimorbidity were obtained based on exhaustive insurance claims data and various types of

**Table 5. Multivariate predictions of the annual total medical cost and 5-year mortality in MetS classes by a generalized linear model (n = 540,615).**

| Variables | Annual total medical cost in 2015* | | | | | 5-year mortality rate** | | |
|---|---|---|---|---|---|---|---|---|
| | Exp (β) | 95% CI | P value | Average marginal effect*** | | Odds ratio | 95% CI | P value |
| | | | | Cost (JPY) [USD] | 95% CI | | | |
| Age group | | | | | | | | |
| 18–29 years | Ref. | | | Ref. | | Ref. | | |
| 30–39 years | 0.90 | (0.87, 0.93) | <0.001 | -91,723 [-764] | (-121,632, -61,813) [-1,014, -515] | 1.73 | (1.12, 2.65) | 0.013 |
| 40–49 years | 0.84 | (0.81, 0.87) | <0.001 | -143,802 [-1,198] | (-172,215, -115,388) [-1,435, -962] | 2.98 | (1.97, 4.51) | <0.001 |
| 50–59 years | 0.81 | (0.78, 0.83) | <0.001 | -174,658 [-1,455] | (-202,836, -146,479) [-1,690, -1,221] | 3.73 | (2.47, 5.64) | <0.001 |
| 60–65 years | 0.79 | (0.77, 0.82) | <0.001 | -189,669 [-1,581] | (-217,922, -161,416) [-1,816, -1,345] | 4.23 | (2.80, 6.38) | <0.001 |
| Sex | | | | | | | | |
| Male | Ref. | | | Ref. | | Ref. | | |
| Female | 0.94 | (0.93, 0.95) | <0.001 | -45,110 [-376] | (-50,757, -39,463) [-423, -329] | 0.41 | (0.38, 0.44) | <0.001 |
| MetS class | | | | | | | | |
| 1 | Ref. | | | Ref. | | Ref. | | |
| 2 | 0.97 | (0.95, 0.98) | <0.001 | -23,556 [-196] | (-30,744, -16,367) [-256, -136] | 0.90 | (0.82, 0.99) | 0.023 |
| 3 | 1.00 | (0.98, 1.01) | 0.499 | -2,874 [-24] | (-11,205, 5,457) [-93, 45] | 1.01 | (0.91, 1.12) | 0.807 |
| 4 | 1.05 | (1.04, 1.06) | <0.001 | 33,358 [278] | (24,253, 42,463) [202, 354] | 1.55 | (1.41, 1.70) | <0.001 |
| 5 | 1.23 | (1.21, 1.24) | <0.001 | 153,278 [1,277] | (145,233, 161,322) [1,210, 1,344] | 1.43 | (1.32, 1.55) | <0.001 |
| 6 | 1.25 | (1.23, 1.27) | <0.001 | 169,907 [1,416] | (159,060, 180,755) [1,326, 1,506] | 1.79 | (1.62, 1.98) | <0.001 |
| 7 | 1.16 | (1.13, 1.19) | <0.001 | 109,655 [914] | (91,542, 127,767) [763, 1,065] | 2.47 | (2.16, 2.84) | <0.001 |
| Total number of diagnosis codes out of the list of 68 disease labels | | | | | | | | |
| <10 | Ref. | | | Ref. | | Ref. | | |
| 10–19 | 1.05 | (1.04, 1.06) | <0.001 | 37,716 [314] | (31,395, 44,037) [262, 367] | 1.08 | (1.01, 1.15) | 0.034 |
| 20–29 | 1.12 | (1.10, 1.15) | <0.001 | 86,389 [720] | (69,750, 103,028) [581, 859] | 1.29 | (1.14, 1.47) | <0.001 |
| ≥30 | 1.66 | (1.58, 1.76) | <0.001 | 466,362 [3,886] | (405,715, 527,008) [3,381, 4,392] | 1.54 | (1.25, 1.91) | <0.001 |
| The number of outpatient visits | | | | | | | | |
| <12 | Ref. | | | Ref. | | Ref. | | |
| 12–23 | 0.97 | (0.96, 0.98) | <0.001 | -21,722 [-181] | (-28,850, -14,594) [-240, -122] | 0.87 | (0.82, 0.94) | <0.001 |
| 24–35 | 1.02 | (1.01, 1.03) | <0.001 | 16,640 [139] | (8,789, 24,490) [73, 204] | 0.81 | (0.75, 0.88) | <0.001 |
| ≥36 | 1.22 | (1.20, 1.23) | <0.001 | 152,287 [1,269] | (143,134, 161,441) [1,193, 1,345] | 0.85 | (0.79, 0.93) | <0.001 |
| Length of hospital stay (days) | | | | | | | | |
| <3 | Ref. | | | Ref. | | Ref. | | |
| 3–7 | 1.64 | (1.62, 1.66) | <0.001 | 315,945 [2,633] | (306,401, 325,489) [2,553, 2,712] | 1.00 | (0.92, 1.10) | 0.936 |

(*Continued*)

**Table 5.** (Continued)

| Variables | Annual total medical cost in 2015* | | | | | 5-year mortality rate** | | |
|---|---|---|---|---|---|---|---|---|
| | Exp (β) | 95% CI | P value | Average marginal effect*** | | Odds ratio | 95% CI | P value |
| | | | | Cost (JPY) [USD] | 95% CI | | | |
| 8–30 | 2.59 | (2.56, 2.62) | <0.001 | 782,969 [6,525] | (769,956, 795,982) [6,416, 6,633] | 1.51 | (1.41, 1.63) | <0.001 |
| ≥31 | 5.06 | (4.98, 5.15) | <0.001 | 1,997,999 [16,650] | (1,960,708, 2,035.290) [16,339, 16,961] | 2.79 | (2.57, 3.02) | <0.001 |
| The number of treatments | | | | | | | | |
| <50 | | | | Ref. | | Ref. | | |
| 50–74 | 1.12 | (1.11, 1.13) | <0.001 | 76,450 [637] | (71,310, 81,591) [594, 680] | 0.92 | (0.87, 0.98) | 0.006 |
| 75–99 | 1.23 | (1.21, 1.24) | <0.001 | 147,507 [1,229] | (138,228, 156,786) [1,152, 1,307] | 0.87 | (0.80, 0.95) | 0.002 |
| ≥100 | 1.48 | (1.45, 1.51) | <0.001 | 313,189 [2,610] | (295,398, 330,979) [2,462, 2,758] | 1.21 | (1.09, 1.35) | <0.001 |
| The number of drugs | | | | | | | | |
| <10 | | | | Ref. | | Ref. | | |
| 10–19 | 1.01 | (1.00, 1.01) | 0.196 | 3,650 [30] | (-1,891, 9,191) [-16, 77] | 0.98 | (0.93, 1.05) | 0.618 |
| 20–29 | 1.11 | (1.10, 1.13) | <0.001 | 80,494 [671] | (71,759, 89,230) [598, 744] | 0.95 | (0.88, 1.03) | 0.228 |
| ≥30 | 1.08 | (1.08, 1.09) | <0.001 | 59,508 [496] | (53,499, 65,517) [446, 546] | 1.29 | (1.22, 1.36) | <0.001 |

* A generalized linear model with log-link function and gamma distribution

** Multiple logistic regression analysis

*** 1 USD = 120 JPY

CI, confidence interval; MetS, metabolic syndrome.

diseases collected from health insurance claims data, which provide reliable results from a representative group of Japanese people.

One reason for the different prevalence of multimorbidity is that there is no consensus on the number of comorbidities used to define multimorbidity. Lee et al. demonstrated that the prevalence of multimorbidity defined as two or more comorbidities was estimated to be higher than that defined as three or more comorbidities [16]. Therefore, when comparing the prevalence rates of previous studies, we need to pay attention to the difference in the definition of multimorbidity. Since the present study focused on patients with high medical costs or severe diseases, the prevalence rate of multimorbidity, which was defined as both 2 or more and 3 or more chronic conditions in this study, was higher than that reported in previous studies that included mild patients [17–20]. Anderson et al. showed that 46.3% of the top 10% high-cost patients had 3 or more chronic conditions [21], and Zulman et al. reported that 77% of the top 5% high-cost patients had multimorbidity defined as 3 or more chronic conditions [7]; the proportion of their patients with multimorbidity was smaller than ours defined as 3 or more chronic conditions (91.1%). The prevalence rates of multimorbidity could not be compared between studies because they depend on the method used for evaluating conditions (e.g., self-report, questionnaire, or health insurance claims data). In particular, the prevalence of multimorbidity tends to be estimated higher if the number of diseases used to define the multimorbidity increases [22]. We included multiple chronic conditions (46 diseases), and there was a

possibility that the number of conditions tended to be higher in our study than in other studies. Juul-Larsen et al. recommended including at least the 29 most prevalent chronic conditions when using the chronic condition measurement guide to study multimorbidity [23]. Furthermore, Holzer et al. showed that studies using classifications with under 25 or over 75 chronic conditions tended to yield lower prevalence estimates and proposed choosing a list of chronic conditions that contained 25–75 single conditions [24]. Consequently, it is possible that the prevalence of multimorbidity was underestimated in previous studies that considered a small number of diseases (less than 25) in the analysis.

The present results showed that MetS is particularly important in controlling high medical costs. The prevalence of the MetS class size was 20% among patients in their 30s, but this gradually increased and eventually reached 50% or more after their 50s. The Japanese Ministry of Health, Labour and Welfare introduced a specific health checkup focusing on people older than 40 years of age with MetS in 2008. However, our results suggested that young men in their 30s should also be provided with health education to prevent the aggravation of metabolic disorders. The increase in MetS among women aged approximately 50 years was related to changes in the sex hormone balance associated with menopause [25]. The decrease in estrogen levels in menopausal women is associated with an increased appetite [26], loss of subcutaneous fat, and an increase in intra-abdominal fat, which causes metabolic abnormalities in menopausal women [27]. Although epidemiological studies have suggested an association between osteoarthritis and natural menopause, the causal relationship between sex hormones and osteoarthritis remains unclear [28, 29]. The present results support that obesity [30] and MetS might mediate postmenopausal osteoarthritis. These detailed mechanisms need to be further examined using longitudinal data.

The concept of MetS has been proposed on the basis of pathophysiologies leading to co-occurrences. No studies have examined the actual patterns of MetS complications using real-world data. We identified 7 MetS patterns in Japanese patients with high medical costs. First, patients in MetS class 1 were younger and had fewer comorbidities and lower inpatient costs and 5-year mortality than those in the other MetS classes, indicating that MetS class 1 consisted of patients with mild conditions. MetS class 2 showed almost the same medical cost per person and mortality as MetS class 1 but included some complications characteristic of obesity [31]. In MetS class 3, most patients had motor disorders, and one of the reasons for this might be that muscle dysfunction and mobility impairment are common in patients with long-term complications of diabetes [32]. Conversely, osteoarthritis also causes low physical activity, consequently leading to MetS. It is important for patients with MetS to monitor osteoarthritis early and carefully [33]. MetS classes 4 and 7 included complications in vital organs including the liver and kidney. Evidence linking non-alcoholic fatty liver disease (NAFLD) and CKD has attracted considerable scientific interest in the last decade [34] and our results might support this relationship. However, Akahane et al. reported that NAFLD was not independently associated with CKD in Japanese people [35]. The causality between NAFLD and CKD should be examined in future studies. Finally, MetS classes 5, 6, and 7 showed a higher medical cost per person, higher mortality, and higher rate of inpatients cost due to diseases requiring surgery or many other comorbidities. It is important to control future high medical costs to prevent progression from mild MetS to these critical MetS classes. Herein, we defined the MetS class only if the MetS triad was simultaneously represented, but there were some classes with MetS dyad. Hence, the medical costs associated with metabolic disorders may be higher.

The proportions of mental disease classes were larger among young men and women, and there were 3 types of mental disease classes: mental disease 1, characterized only by mental illness and associated diseases (e.g., schizophrenia, mood disorder, neurotic disorder, and sleep disorder); mental disease 2, and concurrent metabolic diseases such as diabetes and

dyslipidemia in addition to mood disorder and sleep disorder; and mental disease 3, a high proportion of women in addition to mood disorder and sleep disorder. Mental disorders are associated with diabetes [36], with a higher prevalence among women than men [37]. A recent study showed that insulin resistance positively predicted incident major depressive disorder [38], and depression was reported to be a risk factor for diabetes in a meta-analysis [39]. Furthermore, Tong et al. showed that the number of multimorbidities increased with the levels of depression, especially in female, young, and middle-aged individuals [40]. In contrast to mental disease class 1, classes 2 and 3 might have developed secondary to other diseases or sex-specific factors. Total costs in the mental disease classes were less, but retirement rates for 5 years were higher than those in other classes. In Japan, mandatory retirement has been adopted since the age of 60. The median age of each mental disease class was in the 40s, and thus, the high retirement rates in these classes have little to do with retirement age. Mental health intervention among the young to middle-aged populations is also important to maintain labor productivity.

Our results showed that the cost per person in the kidney class was by far the most expensive. One of the reasons for this is the high prevalence of dialysis, which is the second highest in the world [41]. Therefore, prevention of dialysis among young or middle-aged people will be important in future health policies. Furthermore, CKD is associated with multimorbidity [42]. Patients in the kidney disease class had the highest number of comorbidities, and half of them were diagnosed with cardiovascular diseases, which further explains the high medical costs.

To address the issues raised by multimorbidity, it is essential to adopt the concept of integrated care that places patients at the center of their healthcare journey. However, the development of integrated care in Japan is limited compared to other countries as there is an absence of a formal gatekeeping system in the country [43]. Previously, most physicians in Japan underwent postgraduate training in specialized clinical departments, leading to an emphasis on specialization and a shortage of primary care physicians. In recent years, the Ministry of Health, Labour and Welfare has reviewed the postgraduate training system to increase primary care physicians, and the number of primary care physicians in Japan might gradually increase in the future. By increasing the number of primary care physicians and promoting a more interconnected healthcare system, Japanese medical policy can transition towards a comprehensive and patient-focused approach to healthcare delivery.

This study has several limitations. First, because of the cross-sectional design, we could not consider time since diagnosis. Second, severity of illness was not obtained from the health insurance claims database. Third, we could not examine the effect of socioeconomic status on multimorbidity patterns because the present database did not include information on income and educational background. Even in Japan where individuals can access health care at 30% of the total cost, a low-income individual has poorer access to outpatient care and more serious health conditions than their higher income counterparts [44], which might cause higher risk of multimorbidity. Future studies need to examine the other insurance associations that cover individuals who are self-employed as well as other daily wage earners. A low education level has been reported to be associated with an increased risk of multimorbidity [45]; in particular, this trend seems to be stronger among younger generations (less than 55 years) [46]. Further studies including SES or other factors such as lifestyle, clinical variables, and functional variables are necessary to completely understand multimorbidity. Fourth, although visceral obesity is a prerequisite for MetS, the obesity code (E66) was not considered for identification of the MetS class. Fifth, the prevalence rate of E66 in this population was 1.7%, which was considerably low compared with the prevalence of obesity (defined in Japan as a body mass index of 25.0 kg/m$^2$ or more; male individuals, 28.4%; female individuals, 18.7%) [47]. This is related to

the fact that few doctors provide the disease name of obesity when claiming insurance, and there should be more obese patients even if there is no E66. Sixth, we did not include older adults older than 65 years of age for the following reasons. The present database mostly comprises a working age population and includes only 5% of adults older than 65 years of age because the Japanese law stipulates that the employment obligations by enterprises is up to the age of 65. In addition, the co-payment amount is 30% of the total medical cost for those younger than 70 years of age, whereas that is 20% for those older than 70 years of age if their income is not the same as that of the working age population. Kato et al. [48] examined the effects of different co-payments on the utilization of health care by a regression discontinuity design and demonstrated that a lower co-payment increased outpatient expenditure. Because of this, an increase in healthcare service utilization might be induced by not only multimorbidity but also the co-payment model in adults older than 70 years of age in the present database.

## Conclusion

Most working-age patients with high medical costs suffered from multimorbidity in Japan. Multimorbidity patterns leading to high medical costs per person were MetS, malignancy, and kidney disease classes. Considering the large number of patients and high total medical expenses, dealing with MetS is particularly important to control the high medical costs. In particular, it is necessary to prevent the aggravation of MetS comorbidities, such as cardiovascular diseases or malignancies, in Japan's future medical policy.

## Supporting information

**S1 Checklist. Strengthening the reporting of observational studies in epidemiology statement—A checklist of items that should be included in reports of observational studies.**
(DOC)

**S1 Fig. Distribution of annual medical costs in the insured population who were subscribers in 2015 (n = 16,989,029).** The top 1%, 5%, 10%, 20%, 30% of patients accounted for 26.1%, 46.9%, 59.0%, 73.9% and 83.3% of the total annual medical costs. 1 USD = 120 JPY.
(DOCX)

**S2 Fig. Model fit for the latent class analysis (BIC and AIC).** BIC = Bayesian Information Criterion; AIC = Akaike Information Criterion.
(DOCX)

**S3 Fig. A 100% stacked vertical bar of the patient count in each class for 20, 25, 30, 35, and 40 latent class models.**
(DOCX)

**S4 Fig. Heatmap for item-response probabilities of disease labels in the 30 latent class models.** Thick red represents a high probability.
(DOCX)

**S1 Table. Prevalences of each ICD10 code among patients with high medical costs.**
ICD10 = International Statistical Classification of Diseases and Related Health Problems 10th Revision.
(XLSX)

**S2 Table. Forty-six chronic disease labels classified by the ICD10 codes.**
(XLSX)

**S3 Table. Prevalences of disease labels in patients with the top 10% cost and low to medium cost.**
(XLSX)

**S4 Table. Demographic characteristics of the patients with the top 10% cost and low to medium cost.**
(XLSX)

## Acknowledgments

The authors would like to thank Takehiko Baba and Yosuke Ihara for their technical assistance and knowledge of insurance societies. We are also grateful to Nobuki Ando, the chairman of the Japan Health Insurance Association, and all the relevant staff for providing the opportunity to conduct this study.

## Author Contributions

**Conceptualization:** Yuki Nishida, Tatsuhiko Anzai, Kunihiko Takahashi, Takahide Kozuma, Eiichiro Kanda, Keita Yamauchi, Fuminori Katsukawa.

**Data curation:** Yuki Nishida, Tatsuhiko Anzai.

**Formal analysis:** Yuki Nishida, Tatsuhiko Anzai.

**Funding acquisition:** Kunihiko Takahashi, Eiichiro Kanda, Keita Yamauchi, Fuminori Katsukawa.

**Methodology:** Yuki Nishida, Tatsuhiko Anzai, Keita Yamauchi.

**Project administration:** Fuminori Katsukawa.

**Supervision:** Kunihiko Takahashi, Keita Yamauchi, Fuminori Katsukawa.

**Writing – original draft:** Yuki Nishida.

**Writing – review & editing:** Tatsuhiko Anzai, Kunihiko Takahashi, Takahide Kozuma, Eiichiro Kanda, Keita Yamauchi, Fuminori Katsukawa.

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
