## [Decision Letter · Decision Letter 0]

26 May 2023

PONE-D-23-01730Multimorbidity patterns in working age population with the top 10%-medical cost from exhaustive insurance claims data of Japan Health Insurance AssociationPLOS ONE

Dear Dr. Katsukawa,

Thank you for submitting your manuscript to PLOS ONE. After careful consideration, we feel that it has merit but does not fully meet PLOS ONE’s publication criteria as it currently stands. Therefore, we invite you to submit a revised version of the manuscript that addresses the points raised during the review process.

We look forward to receiving your revised manuscript.

Kind regards,

Edward Zimbudzi

Academic Editor

PLOS ONE

Journal Requirements:

This study was supported by the Japan Health Insurance Association.

FK received funding from the Japan Health Insurance Association. YN was paid from funding from the Japan Health Insurance Association for 1 year from April 2021 to March 2022. The other authors declare no competing interests.

Reviewers' comments:

Reviewer's Responses to Questions

**Comments to the Author**

1. Is the manuscript technically sound, and do the data support the conclusions?

Reviewer #1: Yes

Reviewer #2: Yes

2. Has the statistical analysis been performed appropriately and rigorously? 

Reviewer #1: Yes

Reviewer #2: Yes

3. Have the authors made all data underlying the findings in their manuscript fully available?

Reviewer #1: Yes

Reviewer #2: No

4. Is the manuscript presented in an intelligible fashion and written in standard English?

Reviewer #1: Yes

Reviewer #2: Yes

5. Review Comments to the Author

Reviewer #1: A. General overview:

This study on multimorbidity is both interesting and timely. It estimates the healthcare costs associated with patients who have multiple chronic conditions and identifies patterns of multimorbidity among the top 10% with the highest healthcare expenditures. The study examines both national-level costs and costs per capita. One of the strengths of the study is its use of an exhaustive health insurance database with a very large study population. The findings have important policy implications, particularly in terms of improving health outcomes among the working age population, which is critical for supporting the country's aging population.

The primary constraints of the study are that it is a cross-sectional study and does not take into account time since diagnosis or severity of the illness. The stage of disease to which the calculated costs apply is not well-defined. Moreover, the study population includes only individuals which are formally employed, and only from small and medium sized enterprises. This excludes the informal sector that encompasses around 20% of the workforce in Japan https://www.imf.org/en/Blogs/Articles/2020/04/30/blog043020-a-new-deal-for-informal-workers-in-asia

In addition, individuals who are self-employed, such as street vendors, as well as other daily wage earners, may belong to the most impoverished demographic in the country and face higher susceptibility to multimorbidity and limited access to healthcare. All of these should be further discussed in the limitation.

B. Major comments:

1. First and foremost, the reviewer considers it very important that the author inserts a box in the main text to highlight the clear definition of each class, and what the different classes (especially the numbered classes) entailed and characterized.

2. In line 81, the author wrote: “Multimorbidity is defined as >=2 coexisting conditions in an individual”. It is important to add the word “chronic” (i.e. 2 or more chronic conditions) as per the WHO definition amongst other sources: https://apps.who.int/iris/bitstream/handle/10665/252275/9789241511650-eng.pdf

a. This also affects how the paper categorizes diseases. Some of the included diseases (e.g. certain infections) are often classified in other research, as acute rather than chronic. Moreover, could the author describe the Class “Simple or incidental morbidity”, the class name implies acute conditions?

3. Could the author elaborate more in the Background on what proportion of healthcare cost is borne by the patients? In line 56, the author wrote “partial cost” in reference to this. Moreover as understood from this sentence, the employee-employer each pays half of the employee’s insurance premium. Is the reviewer’s understanding correct? In this case, the costs of healthcare may be much higher, as this study only captures the costs from the provider’s perspective. This should be mentioned in the limitation.

4. In line 138-140, the author wrote: “Labels assigned to the identified latent disease classes were determined with reference to item-response probabilities and treatment content.” In the heatmap (Supplementary Figure S4), suppose that the reviewer understands the above correctly, then shouldn’t 100% of patients with the disease label “7. Lung cancer” respond “yes” to the Class “Lung Cancer”? Perhaps the author could further clarify in the manuscript the process of identifying latent disease classes to enable better understanding.

5. Line 202, “If most patients within the class were women or men, that class was named as a sex-specific disease”. What was the threshold for a class to be considered sex-specific?

6. In Table 4, please provide the respective p-values.

7. Line 251-252/259, it is interesting that the receipts included diagnostics information. Are these the receipts the patients received after making the payment? Could the author elaborate more on what receipt data are?

8. The author may wish to present the equivalent USD amounts in brackets for ease of interpretation for readers.

9. In the discussion, the author emphasized the importance of implementing prevention interventions but did not address the need for a more integrated, patient-centered care system. This is a crucial initiative for the health system to accommodate patients' multiple needs in a cost-effective manner, particularly given the fragmented care system in Japan outlined in the Background section.

C. Minor comments:

1. In line 68, please change “Japan is one of the most aging countries” to “Japan is one of the countries with the highest aging population.”

2. In line 74, the author wrote: “In the United States, a small number of patients spent most of the total medical expenses”. Please consider revising to: “A small proportion of patients in the United States account for the majority of total medical expenditures.”

3. In line 81, please change “>=2” to “two or more”

4. Line 89, please change “was not introduced” to “has not been introduced”.

5. Please use sub-headings in the result section, for enhanced clarity of the flow.

6. Line 125, please add space after the comma “1,2015”.

7. Line 143-144, please change “the frequency rates in the present study” to “the frequency rates of these diseases in the present study”.

8. Line 154, please remove the underscore “and_marginal”.

9. Line 158, please remove the inner brackets “[quartile [Q]1, Q3]”.

10. Line 163, please revise “After excluding the applicable persons” to “After excluding the inapplicable persons”.

11. Line 170, please revise “spent more” to “incurred higher”.

12. The reviewer advises that the titles of figures (page 36) are inserted under their respective figures (page 39-41). Currently they are on different pages. Please also insert titles for all the Supplementary Figures.

13. In Table 2 (page 14), it is highly recommended that the author change “36=<” and “31=<” to “>=36” and “>=31”. Similarly, please make these changes in Table 4 and other places throughout the manuscript where this may have been written.

14. Table 3 (page 17), “drag cost” should be “drug cost”.

15. In line 204, the author uses the term “chronic multimorbidity”, whilst multimorbidity already implies chronic.

16. Line 210, please revise “>90%” to “more than 90%”. Please do this throughout the manuscript, e.g. line 217, etc.

17. Line 230, please repeat in brackets what “MetS triad” includes.

18. In line 260-261, the author wrote: “In particular, many diseases were included when considering that multimorbidity influenced the high prevalence of multimorbidity”.

Please kindly revise this sentence for a better understanding.

19. Line 263, please revise “recommend” to “recommended”.

20. Line 265, please revise “<25 or >75” to “under 25 or over 75”.

21. Line 305, in brackets please define/give examples of “mental illness”.

22. Line 315, please remove ‘years’ in “the age of 60 years”.

23. Line 316, please revise “have nothing to do with” to “have little to do with”.

24. Line 323-324, please consider revising “which were expected to be reasons other than dialysis for the high medical costs” to “which further explains the high medical costs”.

25. Line 338, please revise “Multimorbidity patterns requiring high medical costs” to “Multimorbidity patterns leading to high medical costs”.

26. The author consistently uses the term “frequency rate” throughout the paper, the reviewer assumes the author is referring to “prevalence rate”. The author might consider using this term instead.

27. Please note that the Conclusion section requires a “Conclusion” heading.

28. The author may wish to consider a final round of proofreading, as some of the ideas could be better expressed.

Reviewer #2: This paper presents another thoughtful approach to studying the concentration of healthcare costs in a population without much previously analyzed public data - in this alone it presents a novel contribution, including nearly all adults in Japan during the study period. Latent class analysis is an appropriate and interpretable approach to determining clusters of related conditions; the numbers and group they present are plausible both statistically and biologically.

My primary feedback is that the comparisons to prior studies need to be made just a little more carefully -- for example, in Line 257 the proportion of multimorbidity is considered lower in previous studies of higher cost populations (followed by a discussion of appropriate caveats including data collection method variability, differing lists of chronic conditions), but keep in mind that the definition of multimorbidity can itself differ widely between articles. For example, Ref # 7 reports on multimorbidity affecting 3 or more body systems, and Ref # 20 defines multimorbidity as 3 or more chronic conditions. Meanwhile, the paper at hand defines multimorbidity as 2 or more chronic conditions. Please addend this section to make these differences more clear.

I also wonder why patients > 65 years old were excluded, as this population comprises a significant portion of the healthcare costs in the US and may well also in Japan. Please include some rationale as to why this population was excluded - as most studies of multimorbidity usually include this population.

6. PLOS authors have the option to publish the peer review history of their article (what does this mean?). If published, this will include your full peer review and any attached files.

Reviewer #1: **Yes: **Phuong Bich Tran

Reviewer #2: **Yes: **Usnish Majumdar

---

## [Author Response · Author response to Decision Letter 0]

21 Jul 2023

Responses to the Editor’s Comments

Dear Editor, 

Thank you very much for the opportunity to submit a revised version of our manuscript. We have carefully reviewed the reviewers’ comments and done our best to address their concerns. We hope that you find our work sufficient for publication.

Journal Requirements:

Comment 1: Please ensure that your manuscript meets PLOS ONE's style requirements, including those for file naming. The PLOS ONE style templates can be found at 

Response:

We have ensured that manuscript meets PLOS ONE's style requirements. 

Comment 2: Thank you for stating the following financial disclosure: 

This study was supported by the Japan Health Insurance Association.

Response:

We have added the following statement in the cover letter: This study was supported by the Japan Health Insurance Association. The funders provided the present database and had no role in the study design, data collection and analysis, decision to publish, or preparation of the manuscript.

Comment 3: Thank you for stating the following in the Competing Interests section: 

FK received funding from the Japan Health Insurance Association. YN was paid from funding from the Japan Health Insurance Association for 1 year from April 2021 to March 2022. The other authors declare no competing interests.

Response:

Accordingly, we have added that statement on lines 428–429. 

Comment 4: In your Data Availability statement, you have not specified where the minimal data set underlying the results described in your manuscript can be found. PLOS defines a study's minimal data set as the underlying data used to reach the conclusions drawn in the manuscript and any additional data required to replicate the reported study findings in their entirety. All PLOS journals require that the minimal data set be made fully available. For more information about our data policy, please see http://journals.plos.org/plosone/s/data-availability.

Response:

We have revised our data statement as follows (lines 417–423): “The data used for this study are third party data owned by the Japan Health Insurance Association (https://www.kyoukaikenpo.or.jp.e.ame.hp.transer.com/), and publicly unavailable because of including personal information. Data are available from the Japan Health Insurance Association (https://www.kyoukaikenpo.or.jp.e.ame.hp.transer.com/g7/cat740/sb7210/20210401/) for those study groups that apply for a competitive research grant and receive funding. The authors confirm that the authors did not have any special access or request privileges that other researchers would not have. Contact information is below: Email 99kenkyu.86t@kyoukaikenpo.or.jp.”  

Responses to Reviewer 1’s Comments

Reviewer 1

Comment 1: This study on multimorbidity is both interesting and timely. It estimates the healthcare costs associated with patients who have multiple chronic conditions and identifies patterns of multimorbidity among the top 10% with the highest healthcare expenditures. The study examines both national-level costs and costs per capita. One of the strengths of the study is its use of an exhaustive health insurance database with a very large study population. The findings have important policy implications, particularly in terms of improving health outcomes among the working age population, which is critical for supporting the country's aging population.

Response:

We appreciate your insightful comments and suggestions. We have revised the manuscript in accordance with your comments.

Our responses to your comments are as follows.

Comment 2: The primary constraints of the study are that it is a cross-sectional study and does not take into account time since diagnosis or severity of the illness. The stage of disease to which the calculated costs apply is not well-defined. Moreover, the study population includes only individuals which are formally employed, and only from small and medium sized enterprises. This excludes the informal sector that encompasses around 20% of the workforce in Japan https://www.imf.org/en/Blogs/Articles/2020/04/30/blog043020-a-new-deal-for-informal-workers-in-asia

In addition, individuals who are self-employed, such as street vendors, as well as other daily wage earners, may belong to the most impoverished demographic in the country and face higher susceptibility to multimorbidity and limited access to healthcare. All of these should be further discussed in the limitation.

Response: Thank you for providing these insights.

 As you suggested, time since diagnosis or severity of illness was considered to affect the medical cost. However, the severity of illness was not obtained from the health insurance claims database, and we have added this limitation in the manuscript (lines 367–368): “Second, severity of illness was not obtained from the health insurance claims database.”

 This population database included both formal and informal workers, and we have clarified that information in the manuscript (lines 99–100): “This database consists of approximately 40 million people younger than 75 years of age who are formally or informally employees at small to medium enterprises and their family members.”

 There are several insurance associations in Japan. Individuals who are self-employed are enrolled in the “National Health Insurance,” which is different from the “Japan Health Insurance” in this study population, and they can access healthcare at 30% of the total cost. However, the previous study showed that low-income individuals in Japan have poorer access to outpatient care and more serious health conditions than their higher income counterparts (Fujita, PLOS ONE, 2016), which might have caused the higher risk of multimorbidity. Future studies need to examine the other insurance associations that cover individuals who are self-employed as well as other daily wage earners. We have added this information to the discussion (lines 370-374): “Even in Japan where individuals can access health care at 30% of the total cost, a low-income individual has poorer access to outpatient care and more serious health conditions than their higher income counterparts [44], which might cause higher risk of multimorbidity. Future studies need to examine the other insurance associations that cover individuals who are self-employed as well as other daily wage earners.”

B. Major comments:

Comment 3:. First and foremost, the reviewer considers it very important that the author inserts a box in the main text to highlight the clear definition of each class, and what the different classes (especially the numbered classes) entailed and characterized.

Response: Thank you for your suggestion. We have created a new table that describes the label names and the features of each class. Please refer to Table 3 on page 17 of the revised manuscript.

Comment 4: In line 81, the author wrote: “Multimorbidity is defined as >=2 coexisting conditions in an individual”. It is important to add the word “chronic” (i.e. 2 or more chronic conditions) as per the WHO definition amongst other sources: https://apps.who.int/iris/bitstream/handle/10665/252275/9789241511650-eng.pdf

a. This also affects how the paper categorizes diseases. Some of the included diseases (e.g. certain infections) are often classified in other research, as acute rather than chronic. Moreover, could the author describe the Class “Simple or incidental morbidity”, the class name implies acute conditions?

Response: We have corrected the definition of multimorbidity by adding the word “chronic” (line 67): “Multimorbidity is defined as two or more chronic coexisting conditions in an individual [8], and it is a growing global challenge with substantial effects on individuals, caregivers, and society.” 

The present study aimed to clarify the contribution of multimorbidity among the working-age population with the highest medical cost using latent class analysis. Because multimorbidity is a condition with multiple chronic diseases and, as you suggested, we have selected 46 chronic conditions from 68 diseases in Table 2 and estimated the prevalence of multimorbidity in these high-cost patients. The list of 46 chronic conditions is summarized in S2 Table and we have added the following text to the manuscript (lines 140–144): “Multimorbidity is formally defined as 2 or more chronic conditions [15]; thus, we selected 46 chronic conditions from 68 diseases in Table 2 and estimated the multimorbidity (S2 Table). In chronic diseases, malignancy was grouped into one category, and the following diseases not important in terms of multimorbidity were excluded: oral disease, eye disease, and skin and subcutaneous tissue disease.” After correcting the multimorbidity prevalence, we revised the results and discussion accordingly throughout the manuscript.

Class 1, which was named as “simple or incidental morbidity” in the previous manuscript, probably included patients with acute conditions because open surgery was frequently performed to treat fractures of various body parts such as the finger, hand, knee, clavicle, and so on. However, class 1 did not show any features because the ICD10 code of acute diseases such as fractures was excluded due to their less than 1%, prevalence, which made us hesitate to use “acute condition.” After reaggregating only 46 chronic diseases, patients in class 1 had a median of 1 chronic complication, i.e., class 1 was not a multimorbidity class. To make the class name easier to understand, we have changed the class 1 name to “single chronic condition.” We have modified the class 1 name in the manuscript (lines 201–203): “Class 1 was named as the “single chronic condition.” Patients in this class frequently underwent surgery for any fractures and had only one chronic condition (1 [1, 2]); therefore, they were considered as patients with non-multimorbidity.”

Comment 5: Could the author elaborate more in the Background on what proportion of healthcare cost is borne by the patients? In line 56, the author wrote “partial cost” in reference to this. Moreover as understood from this sentence, the employee-employer each pays half of the employee’s insurance premium. Is the reviewer’s understanding correct? In this case, the costs of healthcare may be much higher, as this study only captures the costs from the provider’s perspective. This should be mentioned in the limitation.

Line 251-252/259, it is interesting that the receipts included diagnostics information. Are these the receipts the patients received after making the payment? Could the author elaborate more on what receipt data are?

Response: You have raised an important question and we should explain the receipt data so that researchers in countries other than Japan can understand them. 

 Japanese calls “health insurance claims” as “receipt,” that is, receipt database means the same as health insurance claims database. We considered that the phrase “receipt database” would confuse the reader, so we have changed it to “health insurance claims database” through the manuscript. Additionally, the diagnostics information is included in health insurance claims data but not in the bill-payment receipt, which patients receive after payment.

 In Japan, the patient’s co-payment at the hospital is 30% of the total direct medical cost and the remaining 70% is covered by the employee’s insurance premium, of which the employee and employer each pays 50%. Our database covers 100% of the total medical cost, and we believe that the present results capture the costs from the public perspective. We have revised the manuscript accordingly (lines 41–44): “Japan uses a universal medical care insurance system in which all citizens subscribe to medical care insurance systems so that everyone receives treatment at 30% of total direct medical costs including the patient’s co-payment while the remaining 70% is covered by the employee’s insurance premium, of which the employee and employer each pay 50% [1].”

Comment 6: In line 138-140, the author wrote: “Labels assigned to the identified latent disease classes were determined with reference to item-response probabilities and treatment content.” In the heatmap (Supplementary Figure S4), suppose that the reviewer understands the above correctly, then shouldn’t 100% of patients with the disease label “7. Lung cancer” respond “yes” to the Class “Lung Cancer”? Perhaps the author could further clarify in the manuscript the process of identifying latent disease classes to enable better understanding.

Response: We agree that we should clarify the process of identifying the latent disease classes. In latent class analysis, a researcher assigns descriptive names to each class according to their dominant features, and thus, the probability of the disease with the same name as label is not necessarily 100%. We decided class names based on the disease with the highest probability in the class and higher probability than the other classes. In addition, we considered the sex proportion in naming the class. We have described the process of identifying the latent disease classes in details (lines 127–129): “Labels assigned to the identified latent disease classes were determined with reference to the highest item-response probability in the class, higher probability than the other classes, treatment content, and sex proportion.”

Comment 7: Line 202, “If most patients within the class were women or men, that class was named as a sex-specific disease”. What was the threshold for a class to be considered sex-specific?

Response: We determined whether the class was sex-specific based on a sex ratio of over 95%. A male-dominated class and 4 female-dominated classes also showed high prevalence of male genital disorders or female genital tract disorders, respectively. Although the breast cancer class consisted mostly of females (98.3%), the prevalence of female genital tract cancer was not substantial and even males can develop breast cancer infrequently; thus, we decided to separate the breast cancer class from the sex-specific disease class. We have clarified the definition of the sex-specific disease class (lines 212–214): “Although the breast cancer class consisted mostly of females (98.3%), the prevalence of female genital tract cancer was not substantial and even males can develop breast cancer infrequently; thus, the breast cancer class was separated from the sex-specific disease class.”

Comment 8: In Table 4, please provide the respective p-values.

Response: Accordingly, we have added the p-values in Table 5 (pages 26–27). Due to the expiration of the license, we were unable to utilize Stata version 17.0 software in the current revision. Therefore, multivariable-adjusted odds ratios (ORs) and their 95% confidence intervals (CIs) were calculated using R 4.3.0 statistical software. Although the 95% confidence intervals have slightly changed, it does not impact the conclusions of the paper. The average marginal effect of cost was calculated using margins package. We have revised the manuscript (lines 154–155): “R 4.3.0 statistical software was used for the other analyses, and average marginal effect of cost was calculated using the margins package.“

Comment 9: The author may wish to present the equivalent USD amounts in brackets for ease of interpretation for readers.

Response: Accordingly, we have added the USD amount, which was translated at the 2015 rate of 1 USD = 120 JPY, where necessary in the manuscript. We have also added the USD amount to the legend of Fig 2. 

Comment 10: In the discussion, the author emphasized the importance of implementing prevention interventions but did not address the need for a more integrated, patient-centered care system. This is a crucial initiative for the health system to accommodate patients' multiple needs in a cost-effective manner, particularly given the fragmented care system in Japan outlined in the Background section.

Response: Thank you for your suggestions. Previously, most physicians in Japan underwent postgraduate training in specialized clinical departments, leading to an emphasis on specialization and a shortage of primary care physicians. Therefore, it is necessary to increase the number of primary care physicians in the future medical policy. In recent years, the Ministry of Health, Labour and Welfare has reviewed the postgraduate training system to increase primary care physicians, and the number of primary care physicians in Japan might gradually increase in the future. We have revised the discussion as follows (lines 360–365): “Previously, most physicians in Japan underwent postgraduate training in specialized clinical departments, leading to an emphasis on specialization and a shortage of primary care physicians. Therefore, it is necessary to increase the number of primary care physicians in the future medical policy. In recent years, the Ministry of Health, Labour and Welfare has reviewed the postgraduate training system to increase primary care physicians, and the number of primary care physicians in Japan might gradually increase in the future.”

 

C. Minor comments:

Comment 11: In line 68, please change “Japan is one of the most aging countries” to “Japan is one of the countries with the highest aging population.”

Response: Accordingly, we have changed the manuscript (lines 54–55): “Japan is one of the countries with the highest aging population, and there are several studies on health care costs focusing on the elderly population [2, 3].”

Comment 12: In line 74, the author wrote: “In the United States, a small number of patients spent most of the total medical expenses”. Please consider revising to: “A small proportion of patients in the United States account for the majority of total medical expenditures.”

Response: Accordingly, we have revised the manuscript (lines 60–61): “A small proportion of patients in the United States account for the majority of total medical expenditures [5].”

Comment 13: In line 81, please change “>=2” to “two or more”

Response: Accordingly, we have changed the manuscript (lines 67-68): “Multimorbidity is defined as two or more chronic coexisting conditions in an individual [8], and it is a growing global challenge with substantial effects on individuals, caregivers, and society.”

Comment 14: Line 89, please change “was not introduced” to “has not been introduced”.

Response: Accordingly, we have changed the manuscript (lines 74–76): “This is because research on multimorbidity has not progressed much in Japan compared with other countries, which is related to the fact that a formal gatekeeping system has not been introduced in Japan [12].”

Comment 15: Please use sub-headings in the result section, for enhanced clarity of the flow.

Response: Thank you for your suggestion. We have added 6 sub-headings in the results section (lines 158–266). 

Comment 16: Line 125, please add space after the comma “1,2015”.

Response: Accordingly, we have added the space (lines 113–115): “The exclusion criteria were as follows: insurance affiliates younger than 18 years of age and older than 65 years of age as of April 1, 2015 (n=1,454,866) and those who took out insurance or withdrew in the middle of the 2015 fiscal year (n=2,359,770).”

Comment 17: Line 143-144, please change “the frequency rates in the present study” to “the frequency rates of these diseases in the present study”.

Response: We have changed the sentence as follows (lines 131–135): “The prevalence rates of some disease labels (e.g., oral disease, eye disease, chronic inflammation of the upper and lower airways, and skin and subcutaneous tissue) were high in most classes because the present study focused on patients with high medical costs, and the prevalence rates of these diseases in the present study were higher than those in the general population, including healthy individuals.”

Comment 18: Line 154, please remove the underscore “and_marginal”.

Response: Accordingly, we have removed the underscore (lines 147–149): “The generalized linear model (GLM) with a log-link function and gamma distribution was generated to assess the factors influencing the total cost and marginal effects of the total medical cost [16].”

Comment 19: Line 158, please remove the inner brackets “[quartile [Q]1, Q3]”.

Response: Accordingly, we have removed the inner brackets (line 153): “Representative values are shown as medians [quartile Q1, Q3].”

Comment 20: Line 163, please revise “After excluding the applicable persons” to “After excluding the inapplicable persons”.

Response: Accordingly, we have revised the manuscript (lines 159–160): “After excluding the inapplicable persons, we extracted the top 10% of patients with high medical costs and finally analyzed the data of 1,698,902 patients (671,450 women and 1,027,452 men).”

Comment 21: Line 170, please revise “spent more” to “incurred higher”.

Response: Accordingly, we have revised the manuscript (lines 167–169): “Men incurred higher medical cost per capita [879,930 JPY (7,333 USD)] than women [734,570 JPY (6,121 USD)], whereas the number of outpatient visits for women seemed to be higher than that for men.”

Comment 22: The reviewer advises that the titles of figures (page 36) are inserted under their respective figures (page 39-41). Currently they are on different pages. Please also insert titles for all the Supplementary Figures.

Response: Thank you for the advice. We have added titles for all figures (lines 234, 243–244, and 258–259), including the supporting information.

Comment 23: In Table 2 (page 14), it is highly recommended that the author change “36=<” and “31=<” to “>=36” and “>=31”. Similarly, please make these changes in Table 4 and other places throughout the manuscript where this may have been written.

Response: As per journal guidelines, we have used the appropriate symbol (�) in Tables 2 and 5 (lines 175 and 268, respectively) and S4 Table (Excel file).

Comment 24: Table 3 (page 17), “drag cost” should be “drug cost”.

Response: Accordingly, we have corrected the word in Table 4 in the revised manuscript (line 222).

Comment 25: In line 204, the author uses the term “chronic multimorbidity”, whilst multimorbidity already implies chronic.

Response: You are correct. We have removed “chronic” before “multimorbidity” (lines 207–208): “One of these sex-specific classes included younger women (33 [29, 37] years) who received perinatal care, and the other classes consisted of patients with multimorbidity.”

Comment 26: Line 210, please revise “>90%” to “more than 90%”. Please do this throughout the manuscript, e.g. line 217, etc.

Response: Throughout the manuscript, we have changed “>” or “<” to “more than” or “less than,” respectively. 

Comment 27: Line 230, please repeat in brackets what “MetS triad” includes.

Response: We have added the description of the MetS triad in brackets (lines 250–256): “The notable features of each MetS class were as follows: MetS class 1, only MetS triad (type 2 diabetes, dyslipidemia, and hypertensive disease); MetS class 2, triad plus infection, chronic inflammation of the upper and lower airways, oral disease, gastrointestinal tract disease, and other locomotive diseases; MetS class 3, triad plus motor disorder; MetS class 4, triad plus liver disease and kidney disease; MetS class 5, triad plus cardiovascular diseases; MetS class 6, triad plus other metabolic diseases, cardiovascular diseases, gastrointestinal disease, and other locomotive diseases; and MetS class 7, triad plus complex conditions including malignancies.”

Comment 28: In line 260-261, the author wrote: “In particular, many diseases were included when considering that multimorbidity influenced the high prevalence of multimorbidity”. Please kindly revise this sentence for a better understanding.

Response: We have revised the manuscript as follows (lines 293–295): “In particular, the prevalence of multimorbidity tends to be estimated higher if the number of diseases used to define the multimorbidity increases.”

Comment 29: Line 263, please revise “recommend” to “recommended”.

Response: We have corrected the word (lines 296–298): “Juul-Larsen et al. recommended using at least the 29 most prevalent chronic conditions when using the chronic condition measurement guide to study multimorbidity [24].”

Comment 30: Line 265, please revise “<25 or >75” to “under 25 or over 75”.

Response: We have revised the manuscript (lines 298–300): “Furthermore, Holzer et al. showed that studies using classifications with under 25 or over 75 chronic conditions tended to yield lower prevalence estimates and proposed choosing a list of chronic conditions that contained 25–75 single conditions [25].”

Comment 31: Line 305, in brackets please define/give examples of “mental illness”.

Response: We have revised the manuscript accordingly (lines 338–343): “The proportions of mental disease classes were larger among young men and women, and there were 3 types of mental disease classes: mental disease 1, characterized only by mental illness and associated diseases (e.g., schizophrenia, mood disorder, neurotic disorder, and sleep disorder); mental disease 2, and concurrent metabolic diseases such as diabetes and dyslipidemia in addition to mood disorder and sleep disorder; and mental disease 3, a high proportion of women in addition to mood disorder and sleep disorder.”

Comment 32: Line 315, please remove ‘years’ in “the age of 60 years”.

Response: Accordingly, we have removed “years” (lines 350–351): “In Japan, mandatory retirement has been adopted since the age of 60.”

Comment 33: Line 316, please revise “have nothing to do with” to “have little to do with”.

Response: As you suggested, “nothing” was an exaggeration in this sentence. We have corrected the manuscript accordingly (lines 351–352): “The median age of each mental disease class was in the 40s, and thus, the high retirement rates in these classes have little to do with retirement age.”

Comment 34: Line 323-324, please consider revising “which were expected to be reasons other than dialysis for the high medical costs” to “which further explains the high medical costs”.

Response: Thank you for the suggestion. We have revised the manuscript accordingly (lines 357–359): “Patients in the kidney disease class had the highest number of comorbidities, and half of them were diagnosed with cardiovascular diseases, which further explains the high medical costs.”

Comment 35: Line 338, please revise “Multimorbidity patterns requiring high medical costs” to “Multimorbidity patterns leading to high medical costs”.

Response: Accordingly, we have revised the manuscript (lines 395–397): “Most working-age patients with high medical costs suffered from multimorbidity in Japan. Multimorbidity patterns leading to high medical costs per person were MetS, malignancy, and kidney disease classes.”

Comment 36: The author consistently uses the term “frequency rate” throughout the paper, the reviewer assumes the author is referring to “prevalence rate”. The author might consider using this term instead.

Response: Thank you for your suggestion. We have revised “frequency rate” to “prevalence rate” and clearly indicated that the prevalence in the present study means the rate of the diagnosed disease (lines 105–107): “Diagnosed disease prevalence was represented by the International Statistical Classification of Diseases and Related Health Problems 10th Revision (ICD10), which was documented between April 2015 and March 2016.”

Comment 37: Please note that the Conclusion section requires a “Conclusion” heading.

Response: We have added the “Conclusion” heading (line 394).

Comment 38: The author may wish to consider a final round of proofreading, as some of the ideas could be better expressed.

Response: We have done a final round of English proofreading. 

 

Responses to Reviewer 2’s Comments

Comment 1: This paper presents another thoughtful approach to studying the concentration of healthcare costs in a population without much previously analyzed public data - in this alone it presents a novel contribution, including nearly all adults in Japan during the study period. Latent class analysis is an appropriate and interpretable approach to determining clusters of related conditions; the numbers and group they present are plausible both statistically and biologically.

Response:

Thank you for the insightful comments and constructive suggestions. We have revised the manuscript as per your comments.

Our responses to your comments are as follows.

Comment 2: My primary feedback is that the comparisons to prior studies need to be made just a little more carefully -- for example, in Line 257 the proportion of multimorbidity is considered lower in previous studies of higher cost populations (followed by a discussion of appropriate caveats including data collection method variability, differing lists of chronic conditions), but keep in mind that the definition of multimorbidity can itself differ widely between articles. For example, Ref # 7 reports on multimorbidity affecting 3 or more body systems, and Ref # 20 defines multimorbidity as 3 or more chronic conditions. Meanwhile, the paper at hand defines multimorbidity as 2 or more chronic conditions. Please addend this section to make these differences more clear.

Response: We appreciate your comment on this point. Please note that we have added the result in relation to reviewer 1's suggestion (comment 4). Multimorbidity is a condition with multiple chronic diseases, and we need to estimate the prevalence of multimorbidity by counting only chronic diseases. We have selected 46 chronic conditions (S2 Table) from 68 diseases in Table 2 and estimated the multimorbidity. 

Even after multimorbidity was defined as 3 or more chronic conditions, its prevalence (91.1%) was higher than that in previous studies (reference numbers 7 and 20 in the original manuscript), which also defined multimorbidity as 3 or more chronic diseases. With the correction of multimorbidity prevalence, we have discussed the effect of multimorbidity’s definition on its prevalence (lines 280–302): “One reason for the different prevalence of multimorbidity is that there is no consensus on the number of comorbidities used to define multimorbidity. Lee et al. demonstrated that the prevalence of multimorbidity defined as two or more comorbidities was estimated to be higher than that defined as three or more comorbidities [17]. Therefore, when comparing the prevalence rates of previous studies, we need to pay attention to the difference in the definition of multimorbidity. Since the present study focused on patients with high medical costs or severe diseases, the prevalence rate of multimorbidity was defined as both 2 or more and 3 or more chronic conditions in this study was higher than that reported in previous studies that included mild patients [18-21]. Anderson et al. showed that 46.3% of the top 10% high-cost patients had 3 or more chronic conditions [22], and Zulman et al. reported that 77% of the top 5% high-cost patients had multimorbidity defined as 3 or more chronic conditions [7]; the proportion of their patients with multimorbidity was smaller than ours defined as 3 or more chronic conditions (91.1%). The prevalence rates of multimorbidity could not be compared between studies because they depend on the method used for evaluating conditions (e.g., self-report, questionnaire, or health insurance claims data). In particular, the prevalence of multimorbidity tends to be estimated higher if the number of diseases used to define the multimorbidity increases [23]. We included multiple chronic conditions (46 diseases), and there was a possibility that the number of conditions tended to be higher in our study than in other studies. Juul-Larsen et al. recommended using at least the 29 most prevalent chronic conditions when using the chronic condition measurement guide to study multimorbidity [24]. Furthermore, Holzer et al. showed that studies using classifications with under 25 or over 75 chronic conditions tended to yield lower prevalence estimates and proposed choosing a list of chronic conditions that contained 25–75 single conditions [25]. Consequently, it is possible that the prevalence of multimorbidity was underestimated in previous studies that considered a small number of diseases (less than 25) in the analysis.”

Comment 3: I also wonder why patients > 65 years old were excluded, as this population comprises a significant portion of the healthcare costs in the US and may well also in Japan. Please include some rationale as to why this population was excluded - as most studies of multimorbidity usually include this population.

Response: We agree with your suggestion. We excluded adults older than 65 years of age for two reasons as follow. First, the present database mostly comprises the working age population and includes only 5% of adults older than 65 years of age because the Japanese law stipulates that the employment obligations by enterprises is up to the age of 65. Although some people continue to work after 65 years, they might be more active and　healthy than retired people; thus, the results of adults older than 65 years of age obtained from this database are difficult to generalize. Second, the co-payment is 30% of the total medical cost for those younger than 70 years of age, whereas that is 20% in those older than 70 years of age if their income is not the same as that of the working age population. Kato et al. (Eur J Health Econ. 2022) examined the effects of different burden amounts on the utilization of healthcare using a regression discontinuity design; they demonstrated that a lower burden amount increased outpatient expenditure. Because of this, an increase of the medical cost might be affected by not only multimorbidity but the health care system in adults older than 70 years of age in the present database. We would like to examine the effect of multimorbidity on medical cost and focus on the working age population who share 30% of medical expenses equally. 

 We have added the reason why patients older than 65 years of age were excluded in the limitation section (lines 383-392): “Sixth, we did not include older adults older than 65 years of age for the following reasons. The present database mostly comprises a working age population and includes only 5% of adults older than 65 years of age because the Japanese law stipulates that the employment obligations by enterprises is up to the age of 65. In addition, the co-payment amount is 30% of the total medical cost for those younger than 70 years of age, whereas that is 20% for those older than 70 years of age if their income is not the same as that of the working age population. Kato et al. [48] examined the effects of different co-payments on the utilization of health care by a regression discontinuity design and demonstrated that a lower co-payment increased outpatient expenditure. Because of this, an increase of medical cost might be affected by not only multimorbidity but the health care system in adults older than 70 years of age in the present database.”

---

## [Decision Letter · Decision Letter 1]

15 Aug 2023

PONE-D-23-01730R1Multimorbidity patterns in the working age population with the top 10% medical cost from exhaustive insurance claims data of Japan Health Insurance AssociationPLOS ONE

Dear Dr. Katsukawa,

Thank you for submitting your manuscript to PLOS ONE. After careful consideration, we feel that it has merit but does not fully meet PLOS ONE’s publication criteria as it currently stands. Therefore, we invite you to submit a revised version of the manuscript that addresses the points raised during the review process.

In particular, there are a few well considered additional comments that can potentially enhance the flow and readability of the paper. 

We look forward to receiving your revised manuscript.

Kind regards,

Edward Zimbudzi

Academic Editor

PLOS ONE

Journal Requirements:

Reviewers' comments:

Reviewer's Responses to Questions

**Comments to the Author**

1. If the authors have adequately addressed your comments raised in a previous round of review and you feel that this manuscript is now acceptable for publication, you may indicate that here to bypass the “Comments to the Author” section, enter your conflict of interest statement in the “Confidential to Editor” section, and submit your "Accept" recommendation.

Reviewer #1: All comments have been addressed

Reviewer #2: All comments have been addressed

2. Is the manuscript technically sound, and do the data support the conclusions?

Reviewer #1: Yes

Reviewer #2: Yes

3. Has the statistical analysis been performed appropriately and rigorously? 

Reviewer #1: Yes

Reviewer #2: Yes

4. Have the authors made all data underlying the findings in their manuscript fully available?

Reviewer #1: Yes

Reviewer #2: Yes

5. Is the manuscript presented in an intelligible fashion and written in standard English?

Reviewer #1: No

Reviewer #2: Yes

6. Review Comments to the Author

Reviewer #1: The reviewer acknowledges and appreciates the authors' efforts to incorporate our suggestions in improving the paper. Each suggestion from the reviewer has been carefully considered and addressed by the authors. There are a few additional comments to further enhance the flow and readability of the paper.

Lines 41–44: As the author mentioned in the response to reviewers, only people in the working age population (under the age of 70) co-pay 30% of healthcare cost (older people are entitled to a lower co-payment).

As such, the author could consider slightly rephrasing this to: “Japan implements a universal medical care insurance system in which all citizens subscribe to healthcare insurance systems. People in the working age population (under the age of 70) receives treatment at 30% of total direct medical costs. The remaining 70% is covered by the employee’s insurance premium, of which the employee and employer each pays 50% [1].”

The sentence above has been further revised to improve phrasing and enhance clarity.

Line 67: Please change “chronic coexisting conditions” to “coexisting chronic conditions”

Lines 74–76: “This is because research on multimorbidity has not progressed much in Japan compared with other countries, which is related to the fact that a formal gatekeeping system has not been introduced in Japan.”

Does the author mean “multimorbidity management/care” rather than “research on multimorbidity”?

a. To improve the logical flow, please consider changing to: “The development in multimorbidity management in Japan is limited compared to other countries as there is an absence of a formal gatekeeping system in the country.”

b. Moreover, it is a bit odd that the author mentioned this point here in the introduction without further elaboration. The author may consider moving the above statement to the discussion where the author discussed the need for a more integrated patient-centered care system and the lack of primary care physicians (as in lines 360–365: “Previously, most physicians in Japan underwent postgraduate training in specialized clinical departments, leading to an emphasis on specialization and a shortage of primary care physicians. Therefore, it is necessary to increase the number of primary care physicians in the future medical policy. In recent years, the Ministry of Health, Labour and Welfare has reviewed the postgraduate training system to increase primary care physicians, and the number of primary care physicians in Japan might gradually increase in the future.”)

c. Lines 360–365 as of now seems a bit out of context. The author should consider providing more context and elaboration for this paragraph. This could be achieved by including an opening and closing statement that concisely links the point to the idea of integrated care.

Line 77-78: Please change “emphasized advocating primary prevention” to “emphasized the need for primary preventions”.

Line 100: Please change “formally or informally employees” to “formally or informally employed”.

Line 162-163: “The median number of 46 chronic conditions was 7 [5, 10].”

Is the author attempting to convey that "The median number of co-existing chronic conditions in an individual, out of the selected list of 46 chronic conditions, was 7 [5, 10]"?

If yes, please kindly make these changes here and in Table 2, as well as throughout the manuscript where applicable (e.g. Line 263-264).

Line 180: Please revise “In preliminary analysis” to “In the preliminary analysis”.

Line 215-217: This sentence may be a bit misleading: “The kidney disease class had the smallest number of patients in all classes and showed that more than 90% of individuals had chronic kidney disease (CKD) and hypertension simultaneously.”

As the reviewer understand, more than 90% of individuals who had chronic kidney disease (CKD) also had hypertension simultaneously.

Therefore, the sentence should be revised to reflect this.

Line 230: Please revise “nearly half between” to “nearly half for”.

Line 286: Please revise to: “…multimorbidity, which was defined as both 2 or more and 3 or more chronic conditions in this study, was…”

Line 297: Please revise “recommended using” to “recommended including”.

Line 304: Please revise “The percentage of the MetS class size” to “The prevalence of the MetS class”.

Line 305: Please remove “of the population”.

Line 306: Please change “focused” to “focusing”.

Line 391-392: The author wrote “Because of this, an increase of medical cost might be affected by not only multimorbidity but the health care system…”

The author probably intend to say “Because of this, increase in healthcare service utilization might be induced by not only multimorbidity, but also the co-payment model…”

Reviewer #2: Updated article addresses my comments and the comments of the other reviewers appropriately. I feel this article is appropriate for publication.

7. PLOS authors have the option to publish the peer review history of their article (what does this mean?). If published, this will include your full peer review and any attached files.

Reviewer #1: **Yes: **Phuong Bich Tran

Reviewer #2: **Yes: **Usnish Brandon Majumdar

---

## [Author Response · Author response to Decision Letter 1]

30 Aug 2023

Responses to Reviewer 1’s Comments

Reviewer 1

The reviewer acknowledges and appreciates the authors' efforts to incorporate our suggestions in improving the paper. Each suggestion from the reviewer has been carefully considered and addressed by the authors. There are a few additional comments to further enhance the flow and readability of the paper.

Response: We appreciate your valuable comments and suggestions once again. We have revised the manuscript in accordance with your comments.

Comment 1: Lines 41–44: As the author mentioned in the response to reviewers, only people in the working age population (under the age of 70) co-pay 30% of healthcare cost (older people are entitled to a lower co-payment).

As such, the author could consider slightly rephrasing this to: “Japan implements a universal medical care insurance system in which all citizens subscribe to healthcare insurance systems. People in the working age population (under the age of 70) receives treatment at 30% of total direct medical costs. The remaining 70% is covered by the employee’s insurance premium, of which the employee and employer each pays 50% [1].”

The sentence above has been further revised to improve phrasing and enhance clarity.

Response: Thank you for your suggestion. We have revised the manuscript (lines 41–43). 

Comment 2: Line 67: Please change “chronic coexisting conditions” to “coexisting chronic conditions”

Response: We have revised the manuscript accordingly (line 67).

Comment 3: Lines 74–76: “This is because research on multimorbidity has not progressed much in Japan compared with other countries, which is related to the fact that a formal gatekeeping system has not been introduced in Japan.”

Does the author mean “multimorbidity management/care” rather than “research on multimorbidity”?

a. To improve the logical flow, please consider changing to: “The development in multimorbidity management in Japan is limited compared to other countries as there is an absence of a formal gatekeeping system in the country.”

b. Moreover, it is a bit odd that the author mentioned this point here in the introduction without further elaboration. The author may consider moving the above statement to the discussion where the author discussed the need for a more integrated patient-centered care system and the lack of primary care physicians (as in lines 360–365: “Previously, most physicians in Japan underwent postgraduate training in specialized clinical departments, leading to an emphasis on specialization and a shortage of primary care physicians. Therefore, it is necessary to increase the number of primary care physicians in the future medical policy. In recent years, the Ministry of Health, Labour and Welfare has reviewed the postgraduate training system to increase primary care physicians, and the number of primary care physicians in Japan might gradually increase in the future.”)

c. Lines 360–365 as of now seems a bit out of context. The author should consider providing more context and elaboration for this paragraph. This could be achieved by including an opening and closing statement that concisely links the point to the idea of integrated care.

Response: Thank you for your insightful advice. As the reviewer suggested, we agree with moving the statement ”The development in multimorbidity management in Japan is limited compared to other countries as there is an absence of a formal gatekeeping system in the country” to the discussion. In addition, we have revised the discussion keeping in mind your advice (lines 359-368): “To address the issues raised by multimorbidity, it is essential to adopt the concept of integrated care that places patients at the center of their healthcare journey. However, the development of integrated care in Japan is limited compared to other countries as there is an absence of a formal gatekeeping system in the country [43]. Previously, most physicians in Japan underwent postgraduate training in specialized clinical departments, leading to an emphasis on specialization and a shortage of primary care physicians. In recent years, the Ministry of Health, Labour and Welfare has reviewed the postgraduate training system to increase primary care physicians, and the number of primary care physicians in Japan might gradually increase in the future. By increasing the number of primary care physicians and promoting a more interconnected healthcare system, Japanese medical policy can transition towards a comprehensive and patient-focused approach to healthcare delivery.”

Comment 4: Line 77-78: Please change “emphasized advocating primary prevention” to “emphasized the need for primary preventions”.

Response: We have revised the manuscript accordingly (lines 75-76).

Comment 5: Line 100: Please change “formally or informally employees” to “formally or informally employed”.

Response: We have revised the manuscript accordingly (line 98).

Comment 6: Line 162-163: “The median number of 46 chronic conditions was 7 [5, 10].”

Is the author attempting to convey that "The median number of co-existing chronic conditions in an individual, out of the selected list of 46 chronic conditions, was 7 [5, 10]"?

If yes, please kindly make these changes here and in Table 2, as well as throughout the manuscript where applicable (e.g. Line 263-264).

Response: Yes. We have revised the sentence throughout the manuscript (lines 160-161, Table 2, Table 4, S4 Table). In relation to this revision, we have also modified the sentence “Total number of 68 diseases” to read as “Number of diagnosis codes out of the list of 68 disease labels” (lines 261-263, Table 2, Table 4, Table 5, S4 Table).

Comment 7: Line 180: Please revise “In preliminary analysis” to “In the preliminary analysis”.

Response: We have revised the manuscript accordingly (line 178).

Comment 8: Line 215-217: This sentence may be a bit misleading: “The kidney disease class had the smallest number of patients in all classes and showed that more than 90% of individuals had chronic kidney disease (CKD) and hypertension simultaneously.”

As the reviewer understand, more than 90% of individuals who had chronic kidney disease (CKD) also had hypertension simultaneously.

Therefore, the sentence should be revised to reflect this.

Response: Thank you for your suggestion. We have revised the manuscript as follows (lines 213–215): “The kidney disease class had the smallest number of patients in all classes and showed that the prevalence rates of chronic kidney disease (CKD) and hypertension were 90% and 95%, respectively.”

Comment 8: Line 230: Please revise “nearly half between” to “nearly half for”.

Response: Accordingly, we have revised the manuscript accordingly (line 228).

Comment 9: Line 286: Please revise to: “…multimorbidity, which was defined as both 2 or more and 3 or more chronic conditions in this study, was…”

Response: We have revised the manuscript accordingly (line 285).

Comment 10: Line 297: Please revise “recommended using” to “recommended including”.

Response: We have revised the manuscript accordingly (line 296).

Comment 11: Line 304: Please revise “The percentage of the MetS class size” to “The prevalence of the MetS class”.

Response: We have revised the manuscript accordingly (line 303).

Comment 12: Line 305: Please remove “of the population”.

Response: We have revised the manuscript accordingly (line 304).

Comment 13: Line 306: Please change “focused” to “focusing”.

Response: We have revised the manuscript accordingly (line 305).

Comment 14: Line 391-392: The author wrote “Because of this, an increase of medical cost might be affected by not only multimorbidity but the health care system…”

The author probably intend to say “Because of this, increase in healthcare service utilization might be induced by not only multimorbidity, but also the co-payment model…”

Response: Thank you for your correction. We have revised the manuscript accordingly (lines 394-395).

 

Responses to Reviewer 2’s Comments

Reviewer #2: Updated article addresses my comments and the comments of the other reviewers appropriately. I feel this article is appropriate for publication.

Response: We appreciate your review and feedback. Thank you for considering the article suitable for publication.

---

## [Editor Report · Decision Letter 2]

1 Sep 2023

Multimorbidity patterns in the working age population with the top 10% medical cost from exhaustive insurance claims data of Japan Health Insurance Association

PONE-D-23-01730R2

Dear Dr. Katsukawa,

We’re pleased to inform you that your manuscript has been judged scientifically suitable for publication and will be formally accepted for publication once it meets all outstanding technical requirements.

Kind regards,

Edward Zimbudzi

Academic Editor

PLOS ONE
---

## [Editor Report · Acceptance letter]

18 Sep 2023

PONE-D-23-01730R2 

Multimorbidity patterns in the working age population with the top 10% medical cost from exhaustive insurance claims data of Japan Health Insurance Association 

Dear Dr. Katsukawa:

I'm pleased to inform you that your manuscript has been deemed suitable for publication in PLOS ONE. Congratulations! Your manuscript is now with our production department. 

Kind regards, 

on behalf of

Dr. Edward Zimbudzi 

Academic Editor

PLOS ONE